# The Role of Somatostatin in the Gastrointestinal Tract

**DOI:** 10.3390/biology14050558

**Published:** 2025-05-16

**Authors:** Konstantinos Papantoniou, Ioanna Aggeletopoulou, Ploutarchos Pastras, Christos Triantos

**Affiliations:** Division of Gastroenterology, Department of Internal Medicine, University of Patras, 26504 Patras, Greece; g.papanton@yahoo.gr (K.P.); iaggel@upatras.gr (I.A.); ploutarchosp96@gmail.com (P.P.)

**Keywords:** somatostatin, somatostatin receptors, gastrointestinal tract, D-cells, analogs, clinical practice

## Abstract

Somatostatin is a hormone secreted by a specific type of cell in different parts of the human body. The gastrointestinal tract is its primary source, while it also serves as a key target for this hormone. By interacting with its receptors, somatostatin affects many different functions, including gastrointestinal motility, hormone and enzyme secretion, gastric acid production, and the integrity of the intestinal barrier. These effects have led to the use of somatostatin and its analogs in the treatment of many different medical conditions. However, more studies are needed to determine which patients might additionally benefit from the use of somatostatin analogs in clinical practice and thus improved medical care.

## 1. Introduction

The gastrointestinal (GI) tract is a complex system with multiple functions, including food digestion, absorption of nutrients, and protection from external harmful stimuli through the intestinal barrier. The enteric nervous system (ENS), a complex neuronal network regarded as the third division of the autonomic nervous system (ANS), supports the peristaltic motor, secretory, and immunological functions of the GI tract [1]. The role of the GI tract as an endocrine organ has also been a subject of research for decades. Since secretin, a gastrointestinal hormone produced by the S-cells of the duodenal and jejunal mucosa, was first identified as a substance released upon contact of hydrochloric acid (HCl) with the duodenal mucosa and, to a lesser extent, the jejunal mucosa [2], numerous other gut-derived hormones have been discovered, each playing a crucial role in GI homeostasis and metabolism [3]. The diverse population of enteroendocrine cells (EECs) throughout the GI wall is essential for regulating hormone production in response to a wide range of stimuli [4].

Somatostatin (SST) is a hormone produced by cells in the ANS. Although it is found in the central and peripheral nervous systems, the gut is considered its main production source as well as an important target for SST [5]. The production of SST by delta cells (D-cells), a type of EECs located in the stomach, small intestine, and pancreas, occurs in response to different types of stimuli. SST exerts its effects primarily through interactions with its receptors (SSTRs), mainly inhibiting the production and secretion of other hormones and peptides, such as glucagon, insulin, and growth hormone. Dysregulation of SST metabolism is associated with several clinical complications, while medications targeting SSTRs are used for the treatment of many diseases [6].

In this review, we explore the mechanisms by which SST interacts with SSTRs, and what effects SST-SSTs signaling can have on different parts of the digestive system. We explore the role of SST in diseases affecting the GI tract. Finally, we look into the use of SST analogs (SSTAs) and other molecules targeting SSTs in clinical practice. Through this comprehensive review, we aim to provide evidence related to the importance of SST and its analogs in the GI system.

## 2. Methodology

This review was conducted based on a comprehensive literature search of peer-reviewed articles. Relevant studies were identified through electronic databases including PubMed, Scopus, and Web of Science, using combinations of the following keywords: somatostatin, somatostatin receptors, gastrointestinal tract, hormone secretion, motility, gastric acid, intestinal mucosal barrier, analogs, GI disease. Only articles published in English were included. The inclusion criteria comprised original research articles, reviews, and clinical trials focusing on the physiological and pathophysiological role of somatostatin in the GI tract, as well as its therapeutic applications in the treatment of GI diseases. Articles that were not written in English, lacked relevance to GI functions and diseases, and conference abstracts were not included in the analysis.

## 3. Somatostatin and Its Receptors in the GI Tract

### 3.1. Somatostatin Production and Secretion

SST is a peptide hormone secreted by cells in different areas of the human body. Because of its inhibitory effect on hormone release, SST is also known as growth hormone-inhibiting hormone (GHIH) or somatotropin release-inhibiting factor (SRIF) [7]. SST production is the result of transcription and translation of the *SST* gene, which is located in chromosome 3. Transcription leads to the production of pre-mRNA, which is in turn converted to mature mRNA after processing in the cell nucleus. Translation of mature mRNA results in the production of the precursor protein preprosomatostatin. Cleavage of a signal sequence composed of 24 amino acids results in the formation of prosomatostatin. Further post-translational modifications then take place to create the two naturally occurring forms of SST: somatostatin-14 (SS-14) and somatostatin-28 (SS-28) [8]. The numbers 14 and 28 reflect the amino acid chain length of each SST form. Both isoforms have short half-lives and comparable affinity for SSTRs; however, they are produced in different tissues. SS-14 is mostly found in parts of the central nervous system (CNS), peripheral nerves, and pancreatic D-cells, while SS-28 is mainly expressed in the GI tract [9].

SST secretion is regulated by many different factors, including paracrine factors, hormones and neurotransmitters. The unique characteristics and wide distribution of SST-producing cells in the human body make them potent regulators of hormone release and enable the effects of SST as an inhibitory molecule. In the CNS, SST is produced by GABAergic neurons mainly found in the neocortex and hippocampus [10]. Other such neurons can be found in many different hypothalamic nuclei, where this molecule was first discovered [11]. The presence of SST is also elevated in the ENS; however, D-cells located in the GI mucosa are considered to be the main producers of SST. D-cells are a type of neuroendocrine cell found throughout the GI tract, as well as in the islets of Langerhans in the pancreas [12]. Describing D-cells in gut mucosa is difficult due to their wide distribution among other cell types. The presence of cytoplasmic extensions that terminate near other types of EECs suggests a potential mechanism through which D-cells might influence the secretion of other molecules in the GI tract [13]. Moreover, the open type of D-cells in the GI tract allows them to come in direct contact and interact with contents in the lumen [14]. Pancreatic D-cells have been better described. Their morphology is different from that of alpha and beta cells, with well-defined cell soma and characteristic neurite-like processes that aid paracrine interaction with other cells even at a distance [15]. Adenosine triphosphate (ATP)-sensitive potassium (K_ATP_) channels in the cytoplasmic membranes of D-cells maintain a hyperpolarized membrane potential when open. In response to activating stimuli, these channels close and cause depolarization of the cell membrane. This leads to increased action potential and calcium (Ca^2+^) influx through voltage-gated Ca^2+^ channels, which subsequently leads to SST release from the small secretory granules where it is stored [16].

### 3.2. Somatostatin Receptors in the Gut

SST exerts its effects on different organs by interacting with its receptors. In the 1990s, genes encoding five different types of SSTRs were described in humans and animals, with researchers observing their expression in various tissues [17,18,19]. These genes are located on different human chromosomes and are identified as follows: *SSTR*_1_—Chromosome 14 (14q13); *SSTR*_2_—Chromosome 17 (17q25.1); *SSTR*_3_—Chromosome 22 (22q13.1); *SSTR*_4_—Chromosome 20 (20p11.21); and *SSTR*_5_—Chromosome 16 (16p13.3), according to the IUPHAR Committee on Receptor Nomenclature and Drug Classification subcommittee [20]. Although these genes are located on different human chromosomes and across different species, they all share high nucleotide sequence identity [21]. The presence of introns in the 5′ untranslated region of the *SSTR*_2_ gene makes it unique among other genes encoding SSTRs. Alternative splicing during translation leads to the production of two different SSTR_2_ receptor forms, namely SSTR_2A_ and SSTR_2B_ [22]. Translational regulation of the *SSTR*_2_ gene is a possible way by which various signals influence the effect of SST in different tissues [23].

SSTRs belong to the class A family of G-protein coupled receptors (GPCRs). Zhao et al. described the crystal structure of SSTR_2_ and SSTR_4_ and how its formation changes during the interaction with different ligands. Their characteristic architecture includes seven transmembrane helices, while an additional eighth helix is found parallel to the cytoplasmic membrane. This structure is stabilized by the formation of disulfide bonds between cysteine amino acids. The open conformation of the extracellular part of SSTRs allows their binding with ligands also containing disulfide bonds, such as the two isoforms of SST and SSTAs [24]. This interaction causes the activation of Gi/Go proteins and subsequent inhibition of adenylyl cyclase, which then leads to a reduction in cyclic adenosine monophosphate (cAMP) concentration [25]. SSTRs also cause hyperpolarization of the cytoplasmic membrane and inhibition of Ca^2+^ influx, thus limiting its presence in the cytosol [26]. These molecules act as second messengers that promote the secretion of different proteins, and their downregulation is an effective mechanism by which SST exerts its inhibitory actions on hormone secretion [27]. Other signaling pathways, independent of adenylyl cyclase and Ca^2+^ channels, are also influenced by SSTRs activation, promoting actions such as reduction in cell proliferation, migration, apoptosis and inflammation reduction. However, these pathways are not equally influenced by all SSTR types [28].

The expression of SSTRs has been observed in many different parts of the human and animal GI tract. Shastry et al. confirmed the presence of SSTR_2_ in healthy salivary glands after observing increased uptake of an SSTA with high affinity for the receptor during positron emission tomography (PET) [29]. The presence of SSTRs has been confirmed in neurons of the submucosal and myenteric nervous plexus, both in human and animal models [30,31]. Many studies have shown the presence of SSTR_2_ in parietal and enterochromaffin-like [ECL] cells located in the stomach, where they play an important role in regulating gastric acid production [32]. Emanuilov et al. recently confirmed the expression of SSTR_1_, SSTR_2_, and SSTR_5_ in the small intestine of rats of different ages [33]. Jepsen et al. examined the paracrine interaction between D-cells and glucagon-like peptide-1 (GLP-1) producing L-cells in the small intestine of mice. They concluded that SST reduced GLP-1 secretion mainly by binding to SSTR_2_ and SSTR_5_ on the surface of L-cells [34]. Buscail et al. detected mRNAs responsible for SSTR_2_ production in the large intestine of healthy human subjects using reverse transcription polymerase chain reaction (RT-PCR) [35]. Geltz et al. recently detected the expression of all SSTR types in normal human colonocytes, while investigating the possible role of SST and its receptors in colorectal cancer (CRC) pathogenesis and prognosis [36]. All these studies demonstrate the wide distribution of SSTRs in the gut. The increased presence of SSTRs in different parts of the GI tract allows SST to apply its effects on various cell types and thus influence the function of different parts of the digestive system.

## 4. Effects of Somatostatin on Different GI Functions

The gut is the major producer of, as well as an important target organ for, SST. By interacting with its receptors throughout the GI tract, SST is involved in the regulation of many different functions, including peristalsis, gastric acid, and hormone secretion, as well as the integrity and protective effect of the intestinal barrier. We will focus on the analysis of these SST effects. The important role of SST in the regulation of insulin and glucagon secretion has also been demonstrated in both human and animal models [37,38,39]. As the interaction between different pancreatic cells and the effects of SST in glucose metabolism have been recently analyzed in the literature, viewers are encouraged to extend their reading on other recent articles regarding this topic [40,41].

### 4.1. Motility

GI motility is the result of coordinated contractions of muscular tissues which comprise the outer layers of the gut walls. These muscles are skeletal in the proximal two-thirds of the esophagus and external anal sphincter, allowing for voluntary control, while autonomous activation is characteristic of smooth muscle cells (SMCs), which are present in the rest of the GI tract [42]. Food digestion, nutrient absorption, and waste elimination are mediated by peristaltic movements of the GI tract. Contractile behavior is regulated by many different mechanisms, including hormone secretion [43]. The effect of SST on GI motility has been a subject of investigation for many decades (Figure 1). Peeters et al. found an association between SST and the migrating motor complex (MMC) using manometry [44]. Straathof et al. observed sustained pressure and lack of relaxation of the lower esophageal sphincter after meal ingestion in humans who were receiving SST [45]. These results might be explained by the presence of SSTR_2A_ on the surface of ENS neurons and interstitial cells of Cajal [30]. Inhibition of nitrengic neurons by SST could reduce nitric oxide (NO) production and thus block its relaxing effect on SMCs and LES relaxation [46]. Delayed gastric emptying appears to be mediated by SST and SSTAs through interaction with SSTRs, especially SSTR_3_ [47]. Okamoto et al. found that gastric antral contraction was reduced after octreotide administration in healthy humans [48]. Transit time through the small and large intestines is also regulated by SST. The SST-SSTR_2_ interaction negatively influences the peristalsis of small intestinal SMCs in animal models [49]. SST-positive neurons are present in large concentrations in the intestinal ENS and promote SMC relaxation through acetylcholine release [12]. In the colon, different SSTRs are expressed on circular and longitudinal human colonic muscle layers, with SSTR_2_ being prominent in the former and SSTR_1-3_ found in the latter [50]. Administration of SSTAs increases colonic tonic response and causes prolonged colonic transit time in human and animal models [51,52]. These studies highlight the importance of SST in regulating GI motility.

Figure 1 illustrates how somatostatin (SST) modulates motility throughout the gastrointestinal tract by interacting with specific somatostatin receptors (SSTRs). In the lower esophageal sphincter (LES), SST modulates sphincter tone, predominantly inhibiting nitric oxide (NO)-mediated relaxation, which can lead to increased or reduced LES pressure depending on receptor interactions. In the stomach, SST decreases antral contractions and delays gastric emptying, primarily through SSTR_3_-mediated suppression of excitatory neurotransmitter release. In the small intestine, SST negatively affects the migrating motor complex (MMC) by reducing peristalsis in smooth muscle cells (SMCs) by inhibiting acetylcholine (ACh) release via SSTR_2_ signaling. In the colon, SST increases colonic tone and slows transit. Created with BioRender.com (accessed on 18 March 2025). Abbreviations: SST, somatostatin; SSTR, somatostatin receptor; LES, lower esophageal sphincter; MMC, migrating motor complex; SMCs, smooth muscle cells; ACh, acetylcholine.

### 4.2. Gastric Acid Secretion

The secretion of HCL is one of the major functions of the stomach. The presence of HCL allows for the activation of enzymes involved in digestion, such as pepsin, promoting the absorption of vitamins and essential nutrients in the digestive tract, and has a protective effect against bacteria and other microorganisms [53,54]. Many different cell types present in the stomach wall are involved in gastric acid production and its regulation. They include G cells, which secrete gastrin, D-cells, which produce SST, ECL cells, which produce histamine, and parietal cells which secrete HCL [55]. The regulation of HCl secretion is mainly mediated by the balance between gastrin and SST (Figure 2). During fasting, SST acts on gastrin-secreting G cells and histamine-secreting ECL cells through a paracrine manner to inhibit HCl secretion by interacting with SSTR_2_ on the membrane of these cells [56]. Activation of the vagus nerve during the cephalic phase of digestion promotes gastrin secretion directly through postganglionic neurons in the stomach wall, while it also acts in an indirect way by reducing the secretion of SST by D-cells [57]. These events lead to the release of gastric acid from parietal cells and histamine by ECL cells. Histamine further promotes gastric acid secretion by interacting with H2 receptors on parietal cells and H3 receptors on D-cells [58]. In order to regulate gastric acid production and avoid possible damage due to hyper-acidification, paracrine feedback pathways are activated by increased gastrin concentration and low pH in the stomach. Activation of extrinsic sensory neurons increases SST production and facilitates a return to the basal inter-digestive state, while gastrin also acts through a paracrine pathway to increase SST production and thus decrease its own concentration in the stomach [59,60,61].

The importance of the regulation of gastric acid secretion is evident during infection by *Helicobacter pylori* (Figure 2). *H. pylori* initially survive in the acidic stomach environment through the production of ammonia and pro-inflammatory cytokines, while it also reduces acid secretion by activating calcitonin gene-related peptide (CGRP) extrinsic sensory neurons and aiding SST secretion [62,63]. Chronic *H. pylori* infection can have different effects depending on the colonization site. A decreased presence of G and D-cells has been observed in the stomach of patients during chronic *H. pylori* infection [64,65,66]. Chronic antral infection is associated with increased gastrin secretion and acid production. These patients are prone to the development of duodenal ulcer disease [67,68]. On the other hand, colonization of the body and fundus is associated with inhibition of H^+^-K^+^ ATPase from *H. pylori* products, and subsequent reduction in gastrin and HCL production [69]. Gastrin has a trophic effect on the stomach and promotes cell proliferation, migration, and angiogenesis. The loss of these effects results in atrophic gastritis, a predisposing factor for gastric malignancies [70]. *H. pylori* eradication restores acid secretion; however, other GI diseases, such as gastroesophageal reflux disease (GERD) and Barrett’s esophagus might occur [71,72].

Figure 2 illustrates how somatostatin (SST) regulates gastric acid secretion under normal conditions and during *Helicobacter pylori* (*H. pylori*) infection. In the fasting state, D-cells secrete SST, which binds to somatostatin receptor 2 (SSTR_2_) on G cells to reduce gastrin release and on enterochromaffin-like (ECL) cells to lower histamine output, ultimately decreasing parietal cell stimulation and acid production. Under normal regulation, vagal nerve activation suppresses SST secretion, thereby promoting gastrin and histamine release and increasing acid secretion. In acute *H. pylori* infection, local inflammation and neuroimmune interactions alter D cell function, leading to excessive gastrin release, acid hypersecretion, and subsequent G cell hyperplasia due to prolonged gastrin stimulation. In contrast, chronic corpus/fundus *H. pylori* infection causes loss of parietal and D-cells, resulting in diminished acid secretion and an increased risk of gastric atrophy and intestinal metaplasia. Through these mechanisms, SST plays a crucial role in maintaining acid homeostasis, and disruption of SST-mediated pathways contributes to acid-related pathologies, including peptic ulcer disease and gastric atrophy. Created with BioRender.com (accessed on 18 March 2025). Abbreviations: SST, somatostatin; SSTR_2_, somatostatin receptor 2; G cells, gastrin-secreting cells; D-cells, somatostatin-secreting cells; ECL cells, enterochromaffin-like cells; HCl, hydrochloric acid; *H. pylori*, *Helicobacter pylori*; CGRP, calcitonin gene-related peptide; H2, histamine type 2 receptor.

### 4.3. Regulation of Hormone Secretion and Electrolyte Distribution

The inhibitory effects of SST modulate the release of several molecules in the GI tract (Figure 3). Cholecystokinin (CCK) is a peptide hormone secreted by EECs in the small intestine in response to the presence of lipids and proteins in the lumen. CCK secretion is further stimulated by CCK-releasing peptide (CCK-RP), which is released from intestinal cells [73]. By binding to its receptors (CCKRs), CCK enhances gallbladder contraction, stimulates pancreatic enzyme secretion, and delays gastric emptying [74]. These processes are essential for lipid and protein digestion. CCK also inhibits gastric acid secretion by activating CCKRA, which induces SST release from D-cells [75]. SST reduces gastric acid production; however, several studies indicate that it also inhibits CCK action [76]. Herzig et al. observed reduced pancreatic enzyme secretion due to SST-mediated inhibition of CCK-RP [77]. Similarly, Miyasaka et al. found that administration of the SSTA octreotide to rats decreased both CCK and CCK-RP secretion [78]. In a randomized controlled trial (RCT) including 67 patients who underwent pancreatic surgery, SST administration had a negative trophic effect on exocrine pancreatic cells and granules, thus leading to fewer post-operative side effects [79].

The secretion of a variety of other peptides in the GI tract is downregulated by SST. Peptide Y-Y (PYY) is an anorexigenic substance which is produced by L-cells. These cells are present in the mucosa of the GI tract, mainly in the small and large intestines. PYY reduces appetite and is associated with conditions such as obesity and anorexia nervosa [80]. SST is a potent inhibitor of PYY, with its effects studied in both human and animal models [81,82]. Rigamonti et al. found that PYY concentration decreased after the administration of SST in patients with obesity and those recovering from anorexia nervosa [83]. Ghrelin is a hormone mostly known for its stimulating effect on appetite; however, it is also involved in glucose hemostasis, thermogenesis, muscle differentiation, and bone metabolism and is found in certain types of malignancies [84]. SST regulates ghrelin metabolism through multiple mechanisms. In the CNS, SST interacts with SSTR_2_, leading to increased ghrelin levels following stress-related suppression of food intake and gastric emptying. On the other hand, SST in the GI tract acts in a direct paracrine manner to cause ghrelin reduction, while it also seems to inhibit the expression of ghrelin-O-acyltransferase (GOAT), an enzyme which is pivotal for ghrelin activation [85]. GLP-1 is another peptide secreted by L-cells, mainly in the distal ileum and colon. By interacting with its receptors, it increases insulin secretion from pancreatic beta-cells and promotes weight loss. Moreover, it has a protective cardiovascular and neuronal effect [86]. The presence of SSTRs on L-cells allows SST to regulate GLP-1 secretion. The interaction between GLP-1 and SST has been the subject of many studies. Orgaard et al. observed that GLP_1_’s inhibitory effect on glucagon secretion was reduced following SSTR blockade in rats [87]. Jepsen et al. studied the possible association between SST and GLP-1 in an animal model. They concluded that SST is a potent mediator of GLP-1 secretion, and blocking SSTRs effectively increases the presence of GLP-1 in the gut [34].

SST appears to inhibit anion secretion and promote sodium and chloride absorption in the ileum [88]. Cooke et al. found decreased secretion of chloride (Cl^−^) ions by colonocytes after administering SSTAs in an animal model [89]. Warhurst et al. observed that administration of SST and clonidine resulted in limited production of many second messengers and eventually led to decreased secretion of chloride ions by cells of the colonic mucosa in vitro [90]. Different molecular mechanisms appear to be involved in these processes. SST-SSTR interaction suppresses adenylate cyclase activity and alters the permeability of electrolyte channels, including calcium and potassium channels, in the plasma membrane [91]. The expression of Na^+^/H^+^ (NHE) exchangers is also altered by SST. The increased expression of NHE8 due to the activation of pathways associated with mitogen-activated protein kinase (MAPK) has been observed after SST administration both in human and animal models, resulting in sodium and water reabsorption [92,93]. These studies point to the important contribution of SST to water and electrolyte homeostasis, which are essential for regular intestinal cell function and integrity.

Figure 3 depicts how somatostatin (SST) modulates both exocrine and endocrine processes in the gastrointestinal tract. On the exocrine side, SST inhibits cholecystokinin (CCK) release, thereby reducing pancreatic enzyme secretion. It also downregulates sodium–hydrogen exchanger 3 (NHE3) in the ileum, decreasing sodium and water absorption. On the endocrine side, SST suppresses the secretion of ghrelin by inhibiting its release from enteroendocrine cells or through inhibition of ghrelin O-acyltransferase (GOAT) and diminishes the release of key gut hormones such as CCK, peptide YY (PYY), and glucagon-like peptide-1 (GLP-1). These actions lead to gallbladder contraction, reduced appetite stimulation, delayed gastric emptying, lower insulin secretion, and overall maintenance of metabolic homeostasis. Through these inhibitory effects, SST exerts a critical regulatory influence on digestive function and energy balance. Created with BioRender.com (accessed on 18 March 2025). Abbreviations: SST, somatostatin; CCK, cholecystokinin; NHE3, sodium-hydrogen exchanger 3; GOAT, ghrelin O-acyltransferase; PYY, peptide YY; GLP-1, glucagon-like peptide-1.

### 4.4. Intestinal Mucosal Barrier

The mucosa of the GI tract is a structure where many microorganisms and external substances come into close contact with host cells. The intestinal barrier is primarily composed of gut microbiota, mucus, epithelial cells, immune cells, and their products. Selective barrier permeability is important for nutrient absorption, maintenance of cell integrity and regulation of the host immune system [94]. Many GI peptides are secreted in response to a variety of stimuli that can be found in the intestinal lumen. Some of these peptides, including SST, act on the intestinal barrier components and play a pivotal role in preserving its normal structure and function [88] (Figure 4). The presence of mucus separates intestinal microorganisms from enterocytes and thus forms the first line of defense against possibly harmful stimuli. It is mainly formed by goblet cells which secrete many different molecules, including mucin 2 (MUC2). Song et al. examined the effects of octreotide on mucus production. SST exposure and interaction with SSTR_5_ led to increased MUC2 production through the suppression of the Notch-Hes1 pathway [95]. Other studies indicate that SST might also influence the differentiation of pluripotent stem cells with the involvement of the same pathway, providing a further protective effect on the intestinal barrier [96].

SST also exerts a protective effect on the epithelial component of the intestinal barrier. Epithelial cells are interconnected, and their structure is stabilized by various cytoplasmic and transmembrane proteins that form tight junctions (TJs). Disruption of this structure is associated with many different diseases [97]. SST is involved in the production and regulation of TJs concentration in different tissues, such as keratinocytes and the blood–brain barrier [98,99]. Several studies have examined the effects of SST on epithelial cells and its role in preserving barrier integrity and function in the GI tract. Li et al. showed that the production of claudin-4 and Zonula Occludens 1 (ZO-1), two TJ proteins, is increased after activation of SSTR_5_ and subsequent signaling through the nuclear factor kappa B subunit 1 (NF-κB)—myosin light chain kinase (MLCK)—myosin light chain (MLC) pathway [100]. Cai et al. and Li et al. found increased expression of claudin-4 and ZO-1 after SST injection in mice with colitis. SST downregulated signaling through the ERK_1/2_-MAPK pathway in these models [101,102].

The anti-inflammatory effects of SST have also been investigated in recent years. Casnici et al. observed reduced production of inflammatory cytokines after octreotide administration in an in vitro model of rheumatoid arthritis [103]. Börzsei et al. found increased analgesic and anti-inflammatory effects of an SSTA after interacting with SSTR_4_ in a mouse model [104]. In the gut, SST enhances the immune function of the intestinal barrier through various interactions. Ma et al. reported a dysregulation of the intestinal inflammatory response in SSTR_3_-deficient animals [105]. The increased presence of SST has been positively associated with the number of innate immunity cells found in the GI tract in animal models [106]. Many studies have also focused on the effect of SST on the adaptive immune system in the gut. Peluso et al. reported an inhibitory effect of SST on tumor necrosis factor alpha (TNF-a), interleukin 1b (IL1-β), and 6 (IL6) in human macrophages stimulated by lipopolysaccharide (LPS) in vitro [107]. Chowers et al. observed reduced secretion of several inflammatory molecules in response to SST signaling in human intestinal cells, both in healthy conditions and diseases [108]. Reduced signaling through the toll like receptor 4 (TLR4)–NF-κB pathway and reduced cytokine production was found after SST use in animals with ischemia–reperfusion injury [109]. In a similar animal model, SST reduced the negative effects of intestinal damage and inflammation by enhancing B-cell maturation through the regulation of transcription factors PAX-5 and BLIMP-1 [110]. The presence of SST has been associated with increased secretion of inflammatory factors from activated T-lymphocytes, including IL-2, interferon-gamma, IL-4, and IL-10. These factors influence the immune function of the intestinal barrier [111]. All these studies showcase the important role of SST as a regulator of inflammatory processes in the GI tract.

Figure 4 illustrates the multifaceted role of somatostatin (SST) in regulating the intestinal mucosal barrier proper function. SST enhances mucus layer integrity by promoting MUC2 secretion, potentially via SSTR_5_ activation in goblet cells. It also inhibits the secretion of inflammatory cytokines by immune cells, particularly macrophages, through the inhibition of the NF-κB signaling pathway. Additionally, SST contributes to B-cell maturation and may influence plasma cell activity, thereby modulating immune responses. Furthermore, SST activates the NF-kB-MLCK-MLC pathway, regulating tight junctions and maintaining epithelial barrier function. These combined actions of SST highlight its significant role in maintaining gut homeostasis and modulating immune responses. Created with BioRender.com (accessed on 18 March 2025). Abbreviations: MUC2, mucin 2; SSTR_5_, somatostatin receptor type 5; TLR4, toll-like receptor 4; NF-kB, nuclear factor kappa-light-chain-enhancer of activated B cells; TGF-b, transforming growth factor beta; MLCK, myosin light-chain kinase; MLC, myosin light chain.

## 5. Somatostatin and GI Diseases

As SST influences hormone secretion and functions in different organs, it has long been considered a useful option in the treatment of various clinical conditions. However, SST exhibits a short half-life, making its use in clinical practice very difficult [112]. Considering these limitations, many different SSTAs with longer half-lives have been developed in recent decades, with a few, mainly octreotide, lanreotide, and pasireotide, proving useful in improving patient care [113]. Similarities between SSTs make them susceptible to activation by common types of stimuli, including SSTAs such as octreotide. However, their structural differences and their presence in different tissues have made the development of ligands with better selectivity and specificity a topic for further research. In this section, we examine the use of SSTAs in the treatment of different conditions in the GI tract (Table 1).

### 5.1. Variceal Bleeding

Varices are a common sign of portal hypertension and are frequently encountered in patients with cirrhosis. Common places where varices might develop include the esophagus, stomach, and rectum. Despite recent advances in patient care, acute variceal bleeding (AVB) remains a condition still associated with significant mortality [141]. SSTRs have been found in blood vessels, both in humans and animal models [142,143]. Octreotide is an SSTA which interacts with different SSTRs, but it has a higher affinity for SSTR_2_ [144]. The interaction of octreotide with SSTR_2_ directly promotes vasoconstriction, while also altering the effect of other vasoactive peptides. This results in reduction in portal and variceal pressure, thus aiding the cessation of bleeding [145]. A meta-analysis of 21 Randomized Controllled Trials (RCTs) showed that cirrhotic patients with AVB had similar mortality rates but reduced adverse effects when treated with SSTAs compared with terlipressin and vasopressin [116]. Current European Society of Gastrointestinal Endoscopy (ESGE) guidelines recommend treatment with vasoactive agents for patients that present with suspected AVB for at least 1 to 2 days, with the favorable profile of SSTAs compared to other agents being noted [115]. Results from a recent RCT comparing a 24 h to a 72 h octreotide infusion in patients with esophageal variceal bleeding suggest that a shortened duration of treatment results in comparable patient outcomes, while it might also reduce the length of hospital stay and medical costs [114].

### 5.2. Angiodysplasias

Angiodysplasias are blood vessel malformations which represent a frequent cause of overt and obscure GI bleeding, both in the small and large intestines. They are commonly found in the elderly, and their treatment is often difficult due to the presence of comorbidities and common recurrence of bleeding in many cases [146]. This leads to an increasing requirement for red blood cell and iron transfusions, as well as more frequent hospitalizations for these patients. Current ESGE guidelines recommend endoscopic hemostasis with argon plasma coagulation (APC) as the first-line treatment for angioectasias [147]. However, endoscopic access to the small intestine with balloon enteroscopy is not widely available and often results in incomplete imaging of the small bowel, increasing the likelihood of untreated lesions [148]. Performing frequent endoscopy procedures is not ideal for either patients or doctors, and health care costs also rise without permanent malformation treatment. SST and SSTAs have many different effects on blood vessels, as they promote platelet aggregation, reduce intestinal blood flow and down-regulation of the vascular endothelial growth factor (VEGF) [149]. Their use has been tested in the treatment of bowel angiectasias in addition to APC or as monotherapy. Goltstein et al. performed a meta-analysis of 11 studies and concluded that SSTAs are an effective and safe treatment option for angiodysplasias located in the GI tract, especially in the small bowel and colon [117]. The results of the recent randomized controlled OCEAN trial, which included 62 patients with transfusion dependent angiectasias, showed that long-acting octreotide treatment in addition to standard endoscopic therapy reduces the need for transfusions and frequent endoscopic hemostasis compared to standard endoscopic therapy alone [118]. Results from these studies make SSTAs an attractive, noninvasive treatment choice for GI angiodysplasias.

### 5.3. Gastrointestinal Neuroendocrine Tumors

One of the main therapeutic indications of SSTAs in clinical practice is the treatment of neuroendocrine tumors (NETs). NETs are an uncommon form of neoplasms that are typically composed of neuroendocrine cells. Hormone secretion and carcinoid syndrome are common characteristics of these tumors. The prognosis for these patients is based on cell differentiation and stage at the time of diagnosis [150]. The GI tract is a frequent location where NETs develop. As SSTs are expressed in many NETs, treatment with SSTAs has been an established choice for these patients, especially in cases where tumor resection is difficult [151]. Several randomized trials have shown improved progression-free survival (PFS) and symptom reduction in patients with advanced NETs originating from the GI tract after administration of SSTAs compared to placebo. However, these studies have not found a reduction in overall mortality, possibly due to high crossover rates [119,120,121]. Many patients with NETs present with carcinoid syndrome, a group of symptoms that occur after the direct release of bioactive molecules into the systemic circulation without prior liver metabolism. Treatment with SSTAs appears to reduce the severity of carcinoid syndrome, thus improving the quality of life for these patients [122]. Current guidelines suggest the use of SSTAs as a first-line approach for the treatment of GI NETs with carcinoid syndrome and for tumor growth control in advanced, slowly growing GI NETs with confirmed expression of SSTs [152]. The administration of SSTAs prior to the resection of NETs and possible metastasis to avoid a possible perioperative carcinoid crisis does not appear to effectively prevent this complication [153].

Imaging and therapy of NETs with radiolabeled ligands are very useful in the diagnosis and therapy of these patients. The binding of SSTAs with radiolabelled substances such as 99m-Technetium allows binding to SSTRs and subsequent NET imaging. SSTA use has been applied in single-photon emission computed tomography (SPECT) and PET [154]. Octreotide was the first SSTA used for NET imaging, but newer analogs offer better image quality due to higher receptor affinity and differences in pharmacokinetics [123]. SST scintigraphy allows the detection of patients who are suitable for peptide receptor radionuclide therapy (PPRT), while it also monitors the therapeutic response [124]. PPRT targets tumor cells and delivers radionuclide molecules in a direct manner, resulting in reduced tumor growth and disease progression. Current data suggest that PPRT can improve patient care and quality of life in patients with advanced and metastatic NETs [125].

### 5.4. Treatment and Imaging of Other GI Neoplasias

SST has been associated with many effects that limit tumor development. By interacting with its receptors, SST can directly inhibit cell proliferation, migration, and induce apoptosis [28]. Inhibition of the secretion of hormones and growth factors, such as gastrin and secretin, limitation of angiogenesis, and promotion of vasoconstriction can further restrict the incidence of carcinogenesis [155]. The use of SSTAs has been examined in the treatment of other non-endocrine tumors, including HCC and breast cancer with promising results [156,157,158]. Testing SSTAs in the treatment of other neoplasias originating in the GI tract is therefore a valid option.

SSTRs are expressed in tumor cells in colorectal cancer (CRC). Their presence varies according to tumor type and differentiation [159]. The high expression of SSTR_2_ has been associated with a poor CRC prognosis [160]. The overall limitation of angiogenesis and the anti-inflammatory effects of SST in CRC have been studied. Leiszter et al. found reduced SST concentration in patients with CRC compared to healthy controls, while the application of octreotide induced apoptosis and limited cell proliferation [126]. These effects appear to be mediated through the activation of signaling pathways that reduce tyrosine kinase activity and increase the activity of phosphatases in neoplastic cells [161]. Collucci et al. examined the effect of SST on CRC cells in vitro. The interaction of SST with SSTR_3_ and five limited malignant cell proliferation by suppressing cyclooxygenase-2 (COX-2) [127]. Despite these promising data, clinical trials have failed to demonstrate a significant impact of SSTAs on CRC regression and improvement of overall survival and quality of life [162,163]. Further clinical trials examining the use of SST in CRC using a personalized approach, combining SSTAs with other agents and possible new analogs targeting a wide variety of SSTRs, could provide exciting information and a new direction in CRC treatment [164].

Due to the high expression of SSTRs in many neoplasias of the gut, imaging with labeled SSTAs has been used in clinical practice. Herlin et al. successfully used (99m)Tc-depreotide to perform scintigraphy in 34 patients with esophageal lesions [165]. Kostenich et al. observed increased detection of colon tumors after administration of a fluorescent SSTA and subsequent microscopy and spectrally resolved imaging in a mouse model [128]. However, these methods are expensive and are not widely available, while false positive results in the presence of colonic adenomas have been reported [159,166]. These factors currently make the application of these methods for the detection of GI non-endocrine neoplasias difficult.

### 5.5. Dumping Syndrome

Dumping syndrome is a condition that typically occurs after surgical procedures that influence the anatomy and physiology of the GI tract, most frequently bariatric surgery, vagotomy with pyloroplasty, and esophagectomy [167]. It typically occurs in two phases. In the first phase, patients typically experience symptoms such as nausea, vomiting, abdominal pain, and tachycardia less than an hour after eating a meal. These first symptoms are attributed to the rapid transit of hyperosmolar luminal content in the small intestine. The late phase typically occurs a few hours after ingestion of a meal rich in carbohydrates. Patients may present with fatigue, dizziness, sweating, and flushing attributed to hypoglycemia, after large amounts of insulin are secreted in response to high circulating glucose concentration [168]. Dietary restrictions with low-calorie diets are effective for relieving symptoms; however, they are not always successful despite patient compliance with medical instructions. SST and its analogs suppress gastric emptying, slow transit through the small intestine and affect hormone and electrolyte distribution. They have been tested in the treatment of dumping syndrome [130]. Early studies showed that octreotide injections had a positive effect on symptoms and quality of life in patients with this condition [129,131]. Arts et al. examined the effect of short and long-acting octreotide in 30 patients with postoperative dumping syndrome. They observed reduced symptoms and improved severity scores with both forms, while long-acting octreotide significantly improved the quality of life in these patients [132]. Similar positive effects have been found after pasireotide administration in a more recent phase II study [134]. However, treatment with SSAs also has complications, such as steatorrhea, gallstone formation, and pain at the injection site, which make compliance difficult in many cases. Didden et al. reported that the long-term results of octreotide administration were not as significant as short-term results in 34 patients with refractory dumping syndrome, with many stopping treatment due to limited treatment efficacy or side effects [133]. Wauters et al. also observed increased side effects without a significant impact on the quality of life after lanreotide administration in patients with dumping syndrome in a randomized study [135]. SSTAs are currently used off-label for the treatment of refractory dumping syndrome, with a positive effect being more obvious in the early phase, while clinicians should be alert for possible side effects [167].

### 5.6. Refractory Diarrhea and Digestive Fistulae

Many patients with chronic diarrhea do not respond to typical therapeutic measures, such as antibiotics and non-specific anti-diarrheal agents. Suppression of gastrointestinal motility and pancreatic enzyme secretion by SST and SSTAs makes them a possible treatment option for these patients [169]. Their use has resulted in beneficial effects to patients with different conditions associated with refractory diarrhea, such as familial amyloid polyneuropathy and medullary thyroid carcinoma [136,137]. However, significant heterogeneity among studies does not provide sufficient evidence for the general use of SST analogs as anti-diarrheal agents in clinical practice [91,145]. Diarrhea has also been reported as a side effect of SSTA treatment [169].

Digestive fistulas are a common complication of surgery and inflammatory diseases with variable localization in the GI tract. Their treatment is challenging and high morbidity and mortality rates are observed. Interventions such as bowel resection and parenteral nutrition are frequently required [170]. Reduced GI exocrine secretion and peristaltic movements caused by SSTAs can aid fistula closing. Coughlin et al. systematically reviewed studies regarding the effect of SSTAs on enterocutaneous fistulas. They found that the use of SSTAs as adjuvant therapy decreased the length of hospital stay and fistula healing; however, it did not have a significant impact on mortality [140]. More recent systematic reviews have not identified high-quality evidence supporting the administration of SSTAs for the treatment of post-operative fistulae, despite their frequent use in clinical practice [138,139].

### 5.7. Limitations of Somatostatin Analogs in Clinical Practice

Despite the large number of studies that support the use of SSTAs for the treatment of several diseases affecting the gut, there are factors that limit their frequent use in clinical practice. SST can cause several GI disturbances, such as nausea, vomiting, diarrhea, and constipation. Other common adverse effects include erythema at the injection site, gallstone formation, and a possible negative effect on glucose metabolism [171]. These events make patient compliance difficult and prevent long-term administration of these agents, making estimation of their long-term effects difficult. Moreover, SSTAs are currently used off-label in many cases, as there is a lack of high-quality evidence supporting their administration for the treatment of conditions such as fistulae and refractory diarrhea [139,145]. Further randomized studies are required to further examine the appropriate use of SSTAs in these cases and how to improve long-term compliance. The development of novel SSTAs with reduced rates of adverse reactions could provide exciting new possibilities for future research.

## 6. Study Limitations

This study has certain limitations. Despite the wide selection of studies, the nature of our review was not systematic and we did not keep a record of all studies that were accessed and excluded during our literature search. A more comprehensive form of review might be necessary to provide more evidence-based answers on questions regarding the effects of SST in the GI tract. Moreover, the significant effect of SSΤ on pancreatic islet hormones warrants further discussion. The expression of SSTR2 in α cells and SSTR5 in β cells enables the inhibitory effect of SST on glucagon and insulin secretion [172]. After meal ingestion, the release of SS-28 from the stomach prevents the occurrence of hypoglycemia and may mitigate potential reductions in tissue insulin sensitivity [173]. As SST reduces insulin secretion, the long-acting use of SSTAs can cause hyperglycemia and negatively affect systemic glucose metabolism [174]. This side effect is another reason why clinicians avoid long-term administration of SSTAs. In cases where SSTA treatment is essential, close glucose monitoring along with lifestyle modifications and anti-diabetic treatment can prevent severe complications [175].

## 7. Conclusions

SST is a potent inhibitor of many different processes in the GI tract. Its receptors are spread throughout the digestive system, allowing SST to interact with them in various tissues and have a range of effects. SSTRs are GPCRs which apply their effects through various signaling pathways. SST alters GI motility, while its presence is important for the preservation and regular function of many components of the intestinal barrier. Exocrine and endocrine secretion of various hormones and molecules, including gastric acid secretion in the lumen, are also regulated by SST. SST actions in the GI tract make SSTAs a valid therapeutic option in different conditions. Their use is currently indicated for the treatment of variceal bleeding and GI NETs, but their administration is useful in many more diseases of the GI tract. The increasing use of promising new therapeutic techniques, including PPRT, and the positive effects of SSTAs in various conditions are well-documented. However, future studies are needed to determine which patients will most benefit from their off-label use, thus improving patient care.

## Figures and Tables

**Figure 1 biology-14-00558-f001:**
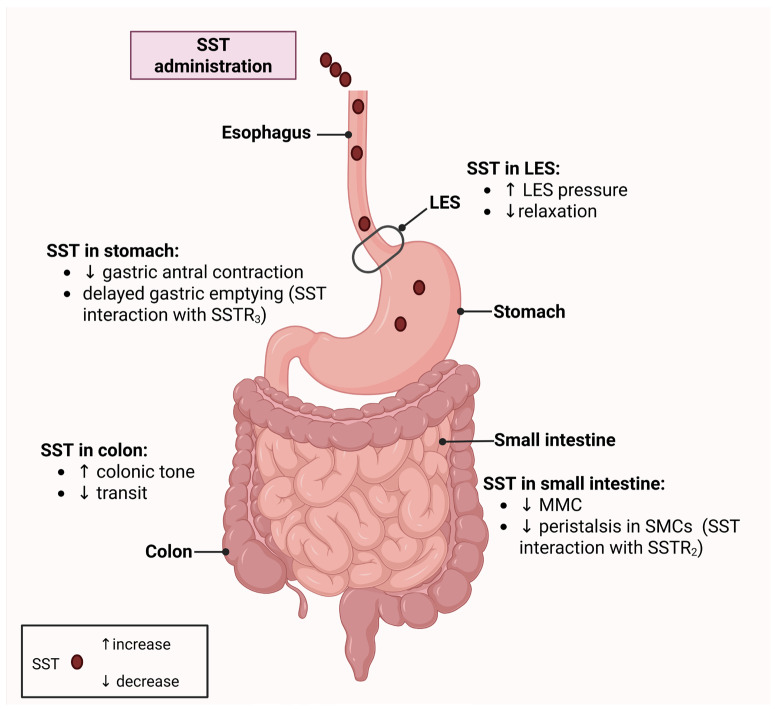
Effects of somatostatin on gastrointestinal motility.

**Figure 2 biology-14-00558-f002:**
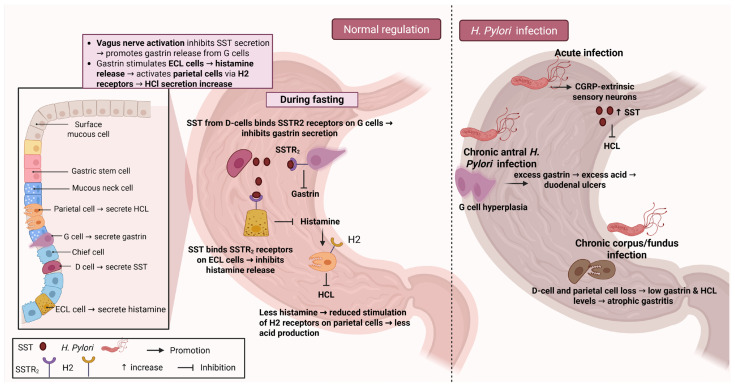
Effects of somatostatin on gastric acid secretion.

**Figure 3 biology-14-00558-f003:**
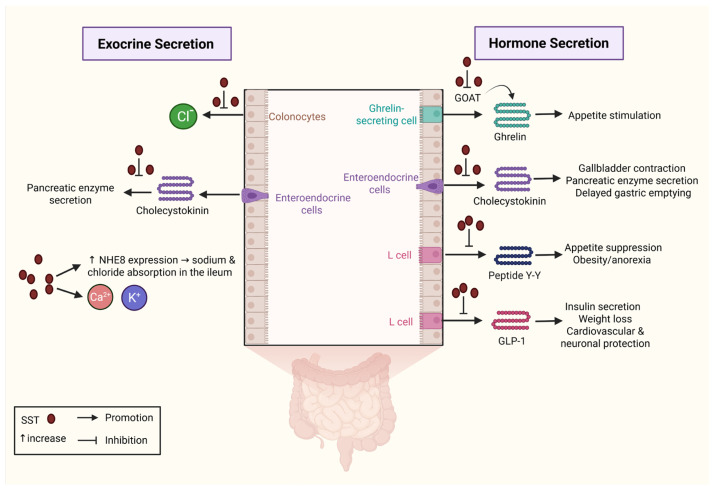
Effects of somatostatin on the regulation of hormone and exocrine secretion.

**Figure 4 biology-14-00558-f004:**
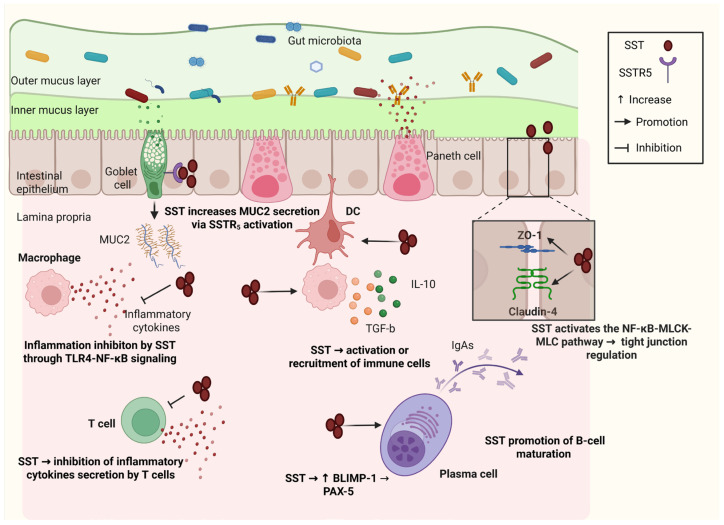
Effects of somatostatin on the regulation of intestinal mucosal barrier integrity.

**Table 1 biology-14-00558-t001:** Use of SSTAs in the treatment of GI diseases.

Medical Condition	SSTAs Effect	Mechanism of Action	On-Label	Studies
Variceal bleeding	Bleeding cessation	Vasoconstriction, portal venous and variceal pressure reduction	Yes	[114,115,116]
GI angiodysplasias	Reduced bleeding rates and transfusion requirements	Platelet aggregation, reduction in intestinal blood flow, down-regulation of VEGF	No	[117,118]
GI NETs	PFS, symptom management, tumor signaling	Reduced hormone secretion, interaction with SSTRs and tumor detection, possibility of PPRT	Yes	[119,120,121,122,123,124,125]
Colorectal cancer	Tumor reduction, tumor signaling	Anti-inflammatory effects, reduced angiogenesis, cell apoptosis, reduced cell proliferation, interaction with SSTRs and tumor detection	No	[126,127,128]
Dumping syndrome	Symptom control, especially in early phase	Slower gastric emptying, slower transit through the small intestine and changes in hormone and electrolyte distribution	No	[129,130,131,132,133,134,135]
Refractory diarrhea	Symptom control	Altered GI motility, reduced exocrine pancreatic secretion	No	[136,137]
Digestive fistulae	Fistula closing	Reduced GI exocrine secretion, altered GI motility	No	[138,139,140]

Abbreviations: SSTAs, somatostatin analogs; GI, gastrointestinal; VEGF, vascular endothelial growth factor; NETs, neuroendocrine tumors; PPRT, peptide receptor radionuclide therapy.

## Data Availability

No new data were created or analyzed in this study. Data sharing is not applicable to this article.

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
