# Peer review of "The Role of Somatostatin in the Gastrointestinal Tract"

_biology, 2025, doi:10.3390/biology14050558_

Round 1

Reviewer 1 Report

Comments and Suggestions for Authors

Article Summary:

The article “The Role of Somatostatin in the Gastrointestinal Tract” examines the functions of somatostatin and its receptors in the gastrointestinal system. The study explores somatostatin’s effects on motility, gastric acid secretion, hormone regulation, and the intestinal mucosal barrier. It also discusses the clinical applications of somatostatin and its analogues (SSTAs) in treating GI diseases, including variceal bleeding, angiodysplasias, neuroendocrine tumors, and dumping syndrome. The article highlights the therapeutic potential of SSTAs while acknowledging challenges, such as limited efficacy in certain conditions and the need for further research. The findings provide a comprehensive understanding of somatostatin’s role in the GI tract and its clinical relevance.

Review:

The manuscript, “The Role of Somatostatin in the Gastrointestinal Tract”, provides comprehensive insights into the physiological roles of somatostatin and its receptors within the gastrointestinal system. The study effectively addresses the various functions of somatostatin, including its impact on motility, gastric acid secretion, hormone regulation, and intestinal barrier function. It also highlights the clinical relevance of somatostatin analogues (SSTAs) in the treatment of several GI diseases. However, a more in-depth discussion of the limitations of SSTAs in clinical practice and a clearer exploration of the challenges regarding long-term efficacy would strengthen the manuscript. Refining the language for greater conciseness and clarity, particularly in relation to the application of SSTAs in specific GI conditions, would improve readability and overall impact. Overall, the study contributes significantly to our understanding of somatostatin’s role in GI health and disease, while providing a valuable perspective on its clinical applications and future research directions.

However, to strengthen the data and ensure clarity, the suggested revisions should be fully addressed. Furthermore, it is recommended that the manuscript is proofread by a specialist company or by a native professional of the English language.

Major revisions

Other observations/concerns

→ The terms must be standardized to SSTR1 - SSTR5, SST1 - SST5, or SSTR1 - SSTR5. I strongly recommend using the SSTR1 - SSTR5 form.

→ All abbreviations presented after the Figures and Tables in Sections 3 and 4 should be omitted, as they are already defined in the text. Leaving them in these sections would be redundant.

Line 59:

Gene names are often written in italics to distinguish them from other elements. For example, SST (line 59). Proteins/peptides encoded by genes, on the other hand, are not written in italics, e.g., SST and SIRT1. This rule was established to facilitate the differentiation between genes (genomic elements) and gene products (proteins). Therefore, I strongly recommend that authors format all gene abbreviations used in the manuscript in italics.

→ Section 2.1 focuses on the production and secretion of SST; however, some information about receptors is already mentioned there. This could be reorganized so that the discussion of receptors occurs exclusively in Section 2.2.

→ The last paragraph of Section 2.2 discusses several studies on the distribution of SSTRs in the gastrointestinal tract; however, it lacks a clear conclusion on how this distribution impacts the hormone’s function.

Minor revisions

Line 28:

Rewrite the sentence for:

“The gastrointestinal (GI) tract is an organ with many different functions...” to “The gastrointestinal (GI) tract is a complex system with multiple functions...”

Lines 33 and 38:

Rewrite the sentence for:

“Since secretin was first described as a substance produced upon contact of hydrochloric acid (HCL) and the duodenal wall [2], it has been found that numerous other hormones are produced in the gut, playing a crucial role in GI homeostasis and metabolism [3]. The presence of many different types of enteroendocrine cells (EECs) in the GI wall mediates the production of many different hormones in response to a wide array of stimuli [4].” to “Since secretin, a gastrointestinal hormone produced by the S-cells of the duodenal and jejunal mucosa, was first identified as a substance released upon contact of hydrochloric acid (HCl) with the duodenal mucosa and, to a lesser extent, the jejunal mucosa [2], numerous other gut-derived hormones have been discovered, each playing a crucial role in GI homeostasis and metabolism [3]. The diverse population of enteroendocrine cells (EECs) throughout the GI wall is essential for regulating hormone production in response to a wide range of stimuli [4].” In this way, the text is presented in a more fluid way.

Lines 43 to 45:

Rewrite the sentence for:

“By interacting with its receptors (SSTRs), SST mainly exerts an inhibitory effect on the production and secretion of other hormones and peptides, such as glucagon, insulin and growth hormone.” to “SST exerts its effects primarily through interactions with its receptors (SSTRs), mainly inhibiting the production and secretion of other hormones and peptides, such as glucagon, insulin, and growth hormone.” In this way, the text is presented in a more fluid way.

Line 46:

Rewrite the sentence for:

“… medical entities, …” to “… clinical complications, …”

Lines 49 and 50:

Rewrite the sentence for:

“We delve into the association of SST with medical diseases that affect the GI tract.” to “We explore the role of SST in diseases affecting the GI tract.”

Lines 50 and 51:

Rewrite the sentence for:

“… Finally, we will look into …” to “… Finally, we look into …”

Lines 56 to 59:

Rewrite the sentence for:

“Due to the inhibitory effect it exerts on the release of other hormones, it is also known as growth hormone-inhibiting hormone (GHIH) or somatotropin release-inhibiting factor (SRIF).” to “Because of its inhibitory effect on hormone release, SST is also known as growth hormone-inhibiting hormone (GHIH) or somatotropin release-inhibiting factor (SRIF).” In this way, the text is presented in a more fluid way.

Line 61:

Rewrite the sentence for:

“... after processing in cell nucleus.” to “... after processing in the cell nucleus.”

Lines 67 and 68:

Rewrite the sentence for:

“Both isoforms have short half-lives and comparable affinity for SSTRs however they are produced in different tissues.” to “Both isoforms have short half-lives and comparable affinity for SSTRs; however, they are produced in different tissues.” In this way, the text is presented in a more fluid way.

Lines 78 and 80:

Rewrite the sentence for:

“D-cells are a type of neuroendocrine cells found throughout the GI tract as well as in the islets of Langerhans in the pancreas.” to “D-cells are a type of neuroendocrine cell found throughout the GI tract, as well as in the islets of Langerhans in the pancreas.” In this way, the text is presented in a more fluid way.

Lines 80 and 81:

Rewrite the sentence for:

“Description of D-cells in gut mucosa is difficult due to their wide distribution among other cell types .” to “Describing D-cells in the gut mucosa is challenging due to their wide distribution among other cell types.” In this way, the text is presented in a more fluid way.

Lines 88 and 90:

Rewrite the sentence for:

“Adenosine triphosphate (ATP)-sensitive potassium (KATP) channels in D-cells cytoplasmic membranes maintain a hyperpolarized membrane potential when they are open.” to “Adenosine triphosphate (ATP)-sensitive potassium (KATP) channels in the cytoplasmic membranes of D-cells maintain a hyperpolarized membrane potential when open.” In this way, the text is presented in a more fluid way.

Lines 97 to 102:

Rewrite the sentence for:

“These genes are based in different human chromosomes however they share many common nucleotide sequences which are found in different species [20]. The latest nomenclature as agreed by the IUPHAR Committee on Receptor Nomenclature and Drug Classification subcommittee classifies the five classes of SST receptors with uppercase letters, namely SST1-SST5 [21].” to “These genes are located on different human chromosomes, identified as follows: SSTR1 – Chromosome 14 (14q13); SSTR2 – Chromosome 17 (17q25.1); SSTR3 – Chromosome 22 (22q13.1); SSTR4 – Chromosome 20 (20p11.21); and SSTR5 – Chromosome 16 (16p13.3), according to the IUPHAR Committee on Receptor Nomenclature and Drug Classification subcommittee [21]. Although these genes are located on different human chromosomes and across different species, they all share high nucleotide sequence identity.” In this way, the text is presented in a more fluid way.

Lines 102 and 105:

Rewrite the sentence for:

“… SSTR2 …” to “… SSTR2 …”

Line 107:

Rewrite the sentence for:

“SSTRs belong in the family of class A G-protein coupled receptors (GPCRs).” to “SSTRs belong to the class A family of G-protein coupled receptors (GPCRs).”

Lines 120 and 121:

Rewrite the sentence for:

“Other signaling pathways independent of adenylyl cyclase and Ca2+ channels are also influenced by SSTRs activation” to “Other signaling pathways, independent of adenylyl cyclase and Ca2+ channels, are also influenced by SSTR activation” In this way, the text is presented in a more fluid way.

Line 144:

Rewrite the sentence for:

“The gut is the major producer as well as an important target organ for SST.” to “The gut is the major producer of, as well as an important target organ for, SST.” In this way, the text is presented in a more fluid way.

Lines 153 and 154:

Rewrite the sentence for:

“Food digestion, nutrient absorption and waste elimination are mediated by peristaltic movements of the GI tract.” to “Food digestion, nutrient absorption, and waste elimination are mediated by the peristaltic movements of the GI tract.” In this way, the text is presented in a more fluid way.

Lines 153 and 154:

Rewrite the sentence for:

“Peeters et al. found an association between SST and the major motor complex (MMC) using manometry [39].” to “Peeters et al. found an association between SST and the migrating motor complex (MMC) using manometry [39].” In this way, the text is presented in a more fluid way.

Line 159:

Rewrite the sentence for:

“… ingestion in human subjects who were …” to “… ingestion in human who were …”

Lines 163 and 164:

Rewrite the sentence for:

“Delayed gastric emptying appears to be mediated by SST and SSTAs after interaction with SSTRs, especially SSTR3 [42].” to “Delayed gastric emptying appears to be mediated by SST and SSTAs through interactions with SSTRs, especially SSTR3 [42].” In this way, the text is presented in a more fluid way.

Line 165:

Rewrite the sentence for:

“… administration in health human subjects [43].” to “… administration in health human [43].”

Lines 166 and 168:

Rewrite the sentence for:

“SST-SST2 interaction has a negative influence on the peristalsis of small intestinal SMCs in animal models [44].” to “The SST-SST2 interaction negatively influences peristalsis in small intestinal SMCs in animal models [44].” In this way, the text is presented in a more fluid way.

Lines 168 and 169:

Rewrite the sentence for:

“SST positive neurons are present in large concentrations in the intestinal ENS, …” to “SST-positive neurons are present in large concentrations in the intestinal ENS, ...”

Lines 173 and 174:

Rewrite the sentence for:

“These studies showcase the important part of SST in regulating GI motility.” to “These studies highlight the important role of SST in regulating GI motility.”

Lines 192 to 195:

Rewrite the sentence for:

“HCL presence allows the activation of enzymes involved in digestion, such as pepsin, and promotes the absorption of vitamins and essential nutrients in the digestive tract, while it also has a protective effect against bacteria and other microorganisms [48,49].” to “HCl presence allows the activation of enzymes involved in digestion, such as pepsin, promotes the absorption of vitamins and essential nutrients in the digestive tract, and has a protective effect against bacteria and other microorganisms [48,49].” In this way, the text is presented in a more fluid way.

Line 198:

Rewrite the sentence for:

“Regulation of HCL” to “The regulation of HCl”

Line 201:

Rewrite the sentence for:

“… HCL ...” to “…. HCl ...”

Line 202:

Rewrite the sentence for:

“Irritation ...” to “ Activation ...”

Line 205:

Rewrite the sentence for:

“… gastric acid from parietal cells and histamine from ECL cells.” to “ … gastric acid by parietal cells and histamine by ECL cells.”

Line 214:

Rewrite the sentence for:

“… the Helicobacter Pylori (HP) (Figure 2.” to “… the Helicobacter Pylori (Figure 2).”

Additionally, HP is not a valid taxonomic abbreviation, so I suggest replacing HP (throughout the section) with H. pylori.

Lines 225 and 226:

Rewrite the sentence for:

“Loss of these effects results in atrophic gastritis, which is a predisposing factor for gastric malignancies [65].” to “The loss of these effects results in atrophic gastritis, a predisposing factor for gastric malignancies [65].” In this way, the text is presented in a more fluid way.

Line 251:

Rewrite the sentence for:

“…influence …” to “… modulate …”

Lines 253 to 255:

Rewrite the sentence for:

“CCK secretion is further promoted by the presence of a CCK releasing peptide (CCK-RP) from intestinal cells [68].” to “CCK secretion is further stimulated by CCK-releasing peptide (CCK-RP), which is produced by intestinal cells [68].” In this way, the text is presented in a more fluid way.

Lines 255 and 257:

Rewrite the sentence for:

“By interacting with its receptors (CCKRs), CCK increases gallbladder contraction, stimulates the secretion of pancreatic enzymes and delays gastric emptying [69]. These events are very important for lipid and protein digestion.” to “By binding to its receptors (CCKRs), CCK enhances gallbladder contraction, stimulates pancreatic enzyme secretion, and delays gastric emptying [69]. These processes are essential for lipid and protein digestion.” In this way, the text is presented in a more fluid way.

Line 258:

Rewrite the sentence for:

“...interacting with CCKRA and causing SST release from D-cells” to “...activating CCKRA, which induces SST release from D-cells”

Lines 259 to 262:

Rewrite the sentence for:

“... many studies show it also negatively affects CCK action [71]. Herzig et al. observed decreased secretion of pancreatic enzymes due to inhibition of CCK-RP by SST [72], while Miyasaka et al. found that CCK and CCK-RP secretion was decreased after administering the SSTA octreotide to rats [73].” to “... several studies indicate that it also inhibits CCK activity [71]. Herzig et al. observed reduced pancreatic enzyme secretion due to SST-mediated inhibition of CCK-RP [72]. Similarly, Miyasaka et al. found that administration of the SSTA octreotide to rats decreased both CCK and CCK-RP secretion [73].” In this way, the text is presented in a more fluid way.

Lines 275 to 278:

Rewrite the sentence for:

“SST regulates ghrelin metabolism through different mechanisms. In the CNS, SST interacts with SSTR2 to cause an increase of ghrelin concentration after stress-related suppression of food intake and gastric emptying.” to “SST regulates ghrelin metabolism through multiple mechanisms. In the CNS, SST interacts with SSTR2, leading to increased ghrelin levels following stress-related suppression of food intake and gastric emptying.” In this way, the text is presented in a more fluid way.

Lines 275 to 278:

Rewrite the sentence for:

“Orgaard et al. observed that the inhibitory effect of GLP-1 on glycagon secretion was diminished after blockage of SSTRs in rats [82].” to “Orgaard et al. observed that GLP1’s inhibitory effect on glucagon secretion was reduced following SSTR blockade in rats [82].” In this way, the text is presented in a more fluid way.

Lines 295, 353, and 499:

Rewrite the sentence for:

“… in vitro ...” to “… in vitro ...”

Lines 296 to 298:

Rewrite the sentence for:

“SST-SSTRs interaction suppresses adenylate cyclase and alters the permeability of electrolyte channels, such as calcium and potassium channels, on the cytoplasmic membrane [86].” to “SST-SSTR interaction suppresses adenylate cyclase activity and alters the permeability of electrolyte channels, including calcium and potassium channels, in the plasma membrane [86].” In this way, the text is presented in a more fluid way.

Line 298:

Rewrite the sentence for:

“Expression of Na+/H+ (NHE)…” to “Expression of Na+/H+ (NHE) …”

Lines 323 to 325:

Rewrite the sentence for:

“The intestinal barrier is a structure mainly comprised of gut microbiota, mucus, epithelial cells and cells of the immune system and their products.” to “The intestinal barrier is primarily composed of gut microbiota, mucus, epithelial cells, immune cells, and their products.” In this way, the text is presented in a more fluid way.

Lines 328 and 329:

Rewrite the sentence for:

“Some of them, including SST, act on the components of the intestinal layer and play a pivotal role in the preservation of it normal structure and function [83]” to “Some of these peptides, including SST, act on the intestinal barrier components and play a pivotal role in preserving its normal structure and function [83]” In this way, the text is presented in a more fluid way.

Lines 333 to 335:

Rewrite the sentence for:

“SST exposure and interaction with SSTR5 resulted in increased MUC2 production due to suppression of the Notch-Hes1 pathway [90].” to “SST exposure and interaction with SSTR5 led to increased MUC2 production through the suppression of the Notch-Hes1 pathway [90].” In this way, the text is presented in a more fluid way.

Lines 338 to 340:

Rewrite the sentence for:

“SST also has a protective effect on the epithelial component of the intestinal barrier. Epithelial cells are connected and their structure is stabilized through a variety of cytoplasmic and transmembrane proteins which form tight junctions (TJs).” to “SST also exerts a protective effect on the epithelial component of the intestinal barrier. Epithelial cells are interconnected, and their structure is stabilized by various cytoplasmic and transmembrane proteins that form tight junctions (TJs).” In this way, the text is presented in a more fluid way.

Lines 343 to 345:

Rewrite the sentence for:

“In the GI tract, several studies have examined the effect of SST on epithelial cells, and pathways that are involved in preserving barrier integrity and function.” to “Several studies have examined the effects of SST on epithelial cells and its role in preserving barrier integrity and function in the GI tract. In this way, the text is presented in a more fluid way.

Line 345:

Rewrite the sentence for:

“... ZO-1, ...” to “… Zonula Occludens-1 (ZO-1), ...”

Lines 349 and 350:

Rewrite the sentence for:

“Signaling via the ERK1/2-MAPK pathway was downregulated by SST in these models [96,97].” to “SST downregulated signaling through the ERK1/2-MAPK pathway in these models [96,97].”

Line 393:

Rewrite the sentence for:

 “… various medical conditions.” to “… various clinical conditions.”

Line 396:

Rewrite the sentence for:

“... in the past decades, with few of them, ...” to “... in recent decades, with a few of them, ...”

Line 398:

Rewrite the sentence for:

“... from ...” to “... by ...”

Line 411:

Rewrite the sentence for

“…is…” to “…remains…”

Line 412:

Rewrite the sentence for

“…human…” to “…humans…”

Lines 414 and 415:

Rewrite the sentence for

“Interaction of octreotide with SSTR2 directly promotes vasoconstriction, while it also alters the effect of other vasoactive peptides.” to “The interaction of octreotide with SSTR2 directly promotes vasoconstriction, while also altering the effect of other vasoactive peptides.” In this way, the text is presented in a more fluid way.

Line 416:

Rewrite the sentence for:

“… 21 RCTs showed” to “… 21 Randomized Controlled Trials (RCTs).”

Line 419:

Rewrite the sentence for:

“ESGE guidelines …” to “European Society of Gastrointestinal Endoscopy (ESGE) guidelines ...”

Line 420:

Rewrite the sentence for:

“...favourable...” to “...favorable...”

Lines 421 and 422:

Rewrite the sentence for:

“...24 hour to a 72 hour…” to “…24-hour to a 72-hour…”

Lines 428 and 431:

Rewrite the sentence for:

“... difficult, due to the presence of comorbidities and common recurrence of bleeding in many cases [141]. This leads to increasing requirements for red blood cell and iron transfusion and more frequent hospitalization for these patients.” to “... difficult due to the presence of comorbidities and the common recurrence of bleeding in many cases [141]. This leads to an increasing requirement for red blood cell and iron transfusions, as well as more frequent hospitalizations for these patients.” In this way, the text is presented in a more fluid way.

Line 432:

Rewrite the sentence for:

“…first line…” to “…the first-line…”

Lines 434 and 435:

Rewrite the sentence for:

“... making the possibility of untreated lesions higher [143].” to “... increasing the likelihood of untreated lesions [143].”

Lines 435 and 436:

Rewrite the sentence for:

“... frequent endoscopy procedures is not ideal for both patients and doctors, while health care costs ...” to “... frequent endoscopic procedures is not ideal for either patients or doctors, and healthcare costs...”

Line 438:

Rewrite the sentence for:

“...aggregation, reduction of intestinal ...” to “... aggregation, reduce intestinal ...”

Lines 451 and 452:

Rewrite the sentence for:

“NETs are an uncommon form of neoplasms which are typically comprised of neuroendocrine cells.” to “NETs are an uncommon form of neoplasms that are typically composed of neuroendocrine cells.” In this way, the text is presented in a more fluid way.

Line 453:

Rewrite the sentence for:

“Prognosis of these ...” to “The prognosis for these...”

Line 454:

Rewrite the sentence for:

“The GI tract is frequent ...” to "The GI tract is a frequent…”

Line 455:

Rewrite the sentence for:

“ …expressed on… “ to “…expressed in…”

Lines 461 and 462:

Rewrite the sentence for:

“... symptoms that take place after the direct release of bioactive molecules in the systemic circulation ...” to “... symptoms that occur after the direct release of bioactive molecules into the systemic circulation ...” In this way, the text is presented in a more fluid way.

Line 464:

Rewrite the sentence for:

 “... syndrome and thus improve quality ...” to “... syndrome, thus improving the quality ...”

Line 465:

Rewrite the sentence for:

“... first line approach...” to “... first-line approach”

Line 466:

Rewrite the sentence for:

  “…slowly-growing GI” to “... slowly growing GI”

Lines 470 and 471:

Rewrite the sentence for:

“... ligands is very useful in the diagnosis and therapy of these patients.” to “... ligands are very useful in the diagnosis and treatment of these patients.”

Lines 484 and 485:

Rewrite the sentence for:

“... inhibit cell proliferation and migration, and induce apoptosis [28].” to “... inhibit cell proliferation, migration, and induce apoptosis [28].”

Line 493:

Rewrite the sentence for:

“... with poor CRC ...” to “... with a poor CRC ...”

Line 518:

“variatric” should likely be “bariatric” (Check this out).

Line 520:

Rewrite the sentence for:

“... complain of symptoms ...” to “... experience symptoms ...”

Line 521:

Rewrite the sentence for:

“... after meal ingestion.” to “... after eating a meal.”

Lines 529 and 530:

Rewrite the sentence for:

 “... small intestine and affect hormone and electrolyte distribution, they have been tested in the treatment of dumping syndrome [125].” to “... small intestine, and affect hormone and electrolyte distribution. They have been tested in the treatment of dumping syndrome [125].” In this way, the text is presented in a more fluid way.

Lines 549 and 551:

Rewrite the sentence for:

“Suppression of gastrointestinal motility and secretion of pancreatic enzymes by SST and SSTAs make them a possible treatment option for these patients [164].” to “Suppression of gastrointestinal motility and pancreatic enzyme secretion by SST and SSTAs makes them a possible treatment option for these patients [164].” In this way, the text is presented in a more fluid way.

Lines 554 and 555:

Rewrite the sentence for:

“...  evidence for general SST analogue application as anti-diarrheal agents in clinical practice [86,164].” to “… evidence for the general use of SST analogues as anti-diarrheal agents in clinical practice [86,164].”

Lines 571 and 572:

Rewrite the sentence for:

“...  many different tissues and cause a variety of effects.” to “... various tissues and produce a range of effects.”

Lines 579 and 582:

Rewrite the sentence for:

“The more frequent use of exciting new therapeutic techniques, including PPRT, and the positive effect of SSTAs in different conditions are well documented, however future studies are needed to determine which patients will mostly benefit from their off-label use, and thus improve patient care.” to “The increasing use of promising new therapeutic techniques, including PPRT, and the positive effects of SSTAs in various conditions are well-documented. However, future studies are needed to determine which patients will most benefit from their off-label use, thus improving patient care.”

Comments on the Quality of English Language

It is recommended that the manuscript be proofread by a specialized company or by a native English professional.

Author Response

We would like to thank you  for helping us improve our manuscript. Here are our responses to your comments.

Comment 1: It also highlights the clinical relevance of somatostatin analogues (SSTAs) in the treatment of several GI diseases. However, a more in-depth discussion of the limitations of SSTAs in clinical practice and a clearer exploration of the challenges regarding long-term efficacy would strengthen the manuscript.

Response 1: A new paragraph titled “5.7. Limitations of somatostatin analogues in clinical practice” has been added to the text and is highlighted in yellow (lines 593-606).

Comment 2: Refining the language for greater conciseness and clarity, particularly in relation to the application of SSTAs in specific GI conditions, would improve readability and overall impact.

Response 2: A thorough re-reading of the text was done to further assess its readability. We made typographical and grammatical corrections throughout our manuscript to improve its clarity.

Comment 3: It is recommended that the manuscript is proofread by a specialist company or by a native professional of the English language.

Response 3: The manuscript has been read by a native speaker of the English language and suggested corrections were made throughout.

Comment 4: The terms must be standardized to SSTR1 - SSTR5, SST1 - SST5, or SSTR1 - SSTR5. I strongly recommend using the SSTR1 - SSTR5 form.

Response 4: The SSTR1 - SSTR5 form has been applied throughout the text.

Comment 5: All abbreviations presented after the Figures and Tables in Sections 3 and 4 should be omitted, as they are already defined in the text. Leaving them in these sections would be redundant.

Response 5: Although the abbreviations presented after the Figures and Tables in Sections 3 and 4 are already described in the text, we believe their addition makes reading and understanding the information included in Figures and Tables easier for readers.

Comment 6: Gene names are often written in italics to distinguish them from other elements. For example, SST (line 59). Proteins/peptides encoded by genes, on the other hand, are not written in italics, e.g., SST and SIRT1. This rule was established to facilitate the differentiation between genes (genomic elements) and gene products (proteins). Therefore, I strongly recommend that authors format all gene abbreviations used in the manuscript in italics.

Response 6: All gene abbreviations used in the manuscript have been formatted in italics.

Comment 7: Section 2.1 focuses on the production and secretion of SST; however, some information about receptors is already mentioned there. This could be reorganized so that the discussion of receptors occurs exclusively in Section 2.2.

Response 7: The sections have been reorganized so that the discussion of receptors occurs exclusively in Paragraph 3.2. The number of the paragraph has been changed due to the addition of a new section titled “2. Methodology”.

Comment 8: The last paragraph of Section 2.2 discusses several studies on the distribution of SSTRs in the gastrointestinal tract; however, it lacks a clear conclusion on how this distribution impacts the hormone’s function.

Response 8: The sentences “All these studies demonstrate the wide distribution of SSTRs in the gut. The increased presence of SSTRs in different parts of the GI tract allows SST to apply its effects on various cell types and thus influence the function of different parts of the digestive system.” have been added  to this paragraph to show  how this distribution impacts the hormone’s function. They are highlighted in yellow (lines 155-158).

Comment 9: Line 28:

Rewrite the sentence for:

“The gastrointestinal (GI) tract is an organ with many different functions...” to “The gastrointestinal (GI) tract is a complex system with multiple functions...”

Response 9: The sentence has been rewritten as suggested and is highlighted in blue (line 28).

Comment 10: Lines 33 and 38:

Rewrite the sentence for:

“Since secretin was first described as a substance produced upon contact of hydrochloric acid (HCL) and the duodenal wall [2], it has been found that numerous other hormones are produced in the gut, playing a crucial role in GI homeostasis and metabolism [3]. The presence of many different types of enteroendocrine cells (EECs) in the GI wall mediates the production of many different hormones in response to a wide array of stimuli [4].” to “Since secretin, a gastrointestinal hormone produced by the S-cells of the duodenal and jejunal mucosa, was first identified as a substance released upon contact of hydrochloric acid (HCl) with the duodenal mucosa and, to a lesser extent, the jejunal mucosa [2], numerous other gut-derived hormones have been discovered, each playing a crucial role in GI homeostasis and metabolism [3]. The diverse population of enteroendocrine cells (EECs) throughout the GI wall is essential for regulating hormone production in response to a wide range of stimuli [4].” In this way, the text is presented in a more fluid way.

Response 10: The sentence has been rewritten as suggested and is highlighted in blue (lines 34-40).

Comment 11: Lines 43 to 45:

Rewrite the sentence for:

“By interacting with its receptors (SSTRs), SST mainly exerts an inhibitory effect on the production and secretion of other hormones and peptides, such as glucagon, insulin and growth hormone.” to “SST exerts its effects primarily through interactions with its receptors (SSTRs), mainly inhibiting the production and secretion of other hormones and peptides, such as glucagon, insulin, and growth hormone.” In this way, the text is presented in a more fluid way.

Response 11: The sentence has been rewritten as suggested and is highlighted in blue (lines 45-48).

Comment 12: Line 46:

Rewrite the sentence for:

“… medical entities, …” to “… clinical complications, …”

Response 12: The sentence has been rewritten as suggested and is highlighted in blue (line 48).

Comment 13: Lines 49 and 50:

Rewrite the sentence for:

“We delve into the association of SST with medical diseases that affect the GI tract.” to “We explore the role of SST in diseases affecting the GI tract.”

Response 13: The sentence has been rewritten as suggested and is highlighted in blue (lines 51-52).

Comment 14: Lines 50 and 51:

Rewrite the sentence for:

“… Finally, we will look into …” to “… Finally, we look into …”

Response 14: The sentence has been rewritten as suggested and is highlighted in blue (line 52).

Comment 15: Lines 56 to 59:

Rewrite the sentence for:

“Due to the inhibitory effect it exerts on the release of other hormones, it is also known as growth hormone-inhibiting hormone (GHIH) or somatotropin release-inhibiting factor (SRIF).” to “Because of its inhibitory effect on hormone release, SST is also known as growth hormone-inhibiting hormone (GHIH) or somatotropin release-inhibiting factor (SRIF).” In this way, the text is presented in a more fluid way.

Response 15: The sentence has been rewritten as suggested and is highlighted in blue (lines 69-70).

Comment 16: Line 61:

Rewrite the sentence for:

“... after processing in cell nucleus.” to “... after processing in the cell nucleus.”

Response 16: The sentence has been rewritten as suggested and is highlighted in blue (line 73).

Comment 17: Lines 67 and 68:

Rewrite the sentence for:

“Both isoforms have short half-lives and comparable affinity for SSTRs however they are produced in different tissues.” to “Both isoforms have short half-lives and comparable affinity for SSTRs; however, they are produced in different tissues.” In this way, the text is presented in a more fluid way.

Response 17: The sentence has been rewritten as suggested and is highlighted in blue (lines 78-80).

Comment 18: Lines 78 and 80:

Rewrite the sentence for:

“D-cells are a type of neuroendocrine cells found throughout the GI tract as well as in the islets of Langerhans in the pancreas.” to “D-cells are a type of neuroendocrine cell found throughout the GI tract, as well as in the islets of Langerhans in the pancreas.” In this way, the text is presented in a more fluid way.

Response 18: The sentence has been rewritten as suggested and is highlighted in blue (lines 90-92).

Comment 19: Lines 80 and 81:

Rewrite the sentence for:

“Description of D-cells in gut mucosa is difficult due to their wide distribution among other cell types.” to “Describing D-cells in the gut mucosa is challenging due to their wide distribution among other cell types.” In this way, the text is presented in a more fluid way.

Response 19: The sentence has been rewritten as suggested and is highlighted in blue (lines 92-93).

Comment 20: Lines 88 and 90:

Rewrite the sentence for:

“Adenosine triphosphate (ATP)-sensitive potassium (KATP) channels in D-cells cytoplasmic membranes maintain a hyperpolarized membrane potential when they are open.” to “Adenosine triphosphate (ATP)-sensitive potassium (KATP) channels in the cytoplasmic membranes of D-cells maintain a hyperpolarized membrane potential when open.” In this way, the text is presented in a more fluid way.

Response 20: The sentence has been rewritten as suggested and is highlighted in blue (lines 100-102).

Comment 21: Lines 97 to 102:

Rewrite the sentence for:

“These genes are based in different human chromosomes however they share many common nucleotide sequences which are found in different species [20]. The latest nomenclature as agreed by the IUPHAR Committee on Receptor Nomenclature and Drug Classification subcommittee classifies the five classes of SST receptors with uppercase letters, namely SST1-SST5 [21].” to “These genes are located on different human chromosomes, identified as follows: SSTR1 – Chromosome 14 (14q13); SSTR2 – Chromosome 17 (17q25.1); SSTR3 – Chromosome 22 (22q13.1); SSTR4 – Chromosome 20 (20p11.21); and SSTR5 – Chromosome 16 (16p13.3), according to the IUPHAR Committee on Receptor Nomenclature and Drug Classification subcommittee [21]. Although these genes are located on different human chromosomes and across different species, they all share high nucleotide sequence identity.” In this way, the text is presented in a more fluid way.

Response 21: The sentence has been rewritten as suggested and is highlighted in blue (lines 109-113 ).

Comment 22: Lines 102 and 105:

Rewrite the sentence for:

“… SSTR2 …” to “… SSTR…”

Response 22: The sentence has been rewritten as suggested and is highlighted in blue (line 116, 119).

Comment 23: Line 107:

Rewrite the sentence for:

“SSTRs belong in the family of class A G-protein coupled receptors (GPCRs).” to “SSTRs belong to the class A family of G-protein coupled receptors (GPCRs).”

Response 23: The sentence has been rewritten as suggested and is highlighted in blue (line 121).

Comment 24: Lines 120 and 121:

Rewrite the sentence for:

“Other signaling pathways independent of adenylyl cyclase and Ca2+ channels are also influenced by SSTRs activation” to “Other signaling pathways, independent of adenylyl cyclase and Ca2+ channels, are also influenced by SSTR activation” In this way, the text is presented in a more fluid way.

Response 24: The sentence has been rewritten as suggested and is highlighted in blue (line 134-136).

Comment 25: Line 144:

Rewrite the sentence for:

“The gut is the major producer as well as an important target organ for SST.” to “The gut is the major producer of, as well as an important target organ for, SST.” In this way, the text is presented in a more fluid way.

Response 25: The sentence has been rewritten as suggested and is highlighted in blue (line 160).

Comment 26: Lines 153 and 154:

Rewrite the sentence for:

“Food digestion, nutrient absorption and waste elimination are mediated by peristaltic movements of the GI tract.” to “Food digestion, nutrient absorption, and waste elimination are mediated by the peristaltic movements of the GI tract.” In this way, the text is presented in a more fluid way.

Response 26: The sentence has been rewritten as suggested and is highlighted in blue (line 174-175).

Comment 27: Lines 153 and 154:

Rewrite the sentence for:

“Peeters et al. found an association between SST and the major motor complex (MMC) using manometry [39].” to “Peeters et al. found an association between SST and the migrating motor complex (MMC) using manometry [39].” In this way, the text is presented in a more fluid way.

Response 27: The sentence has been rewritten as suggested and is highlighted in blue (line 178-179).

Comment 28: Line 159:

Rewrite the sentence for:

“… ingestion in human subjects who were …” to “… ingestion in human who were …”

Response 28: The sentence has been rewritten as suggested and is highlighted in blue (line 180).

Comment 29: Lines 163 and 164:

Rewrite the sentence for:

“Delayed gastric emptying appears to be mediated by SST and SSTAs after interaction with SSTRs, especially SSTR3 [42].” to “Delayed gastric emptying appears to be mediated by SST and SSTAs through interactions with SSTRs, especially SSTR3 [42].” In this way, the text is presented in a more fluid way.

Response 29: The sentence has been rewritten as suggested and is highlighted in blue (line 184-185).

Comment 30: Line 165:

Rewrite the sentence for:

“… administration in health human subjects [43].” to “… administration in health human [43].”

Response 30: The sentence has been rewritten as suggested and is highlighted in blue (line 186-187).

Comment 31: Lines 166 and 168:

Rewrite the sentence for:

“SST-SST2 interaction has a negative influence on the peristalsis of small intestinal SMCs in animal models [44].” to “The SST-SST2 interaction negatively influences peristalsis in small intestinal SMCs in animal models [44].” In this way, the text is presented in a more fluid way.

Response 31: The sentence has been rewritten as suggested and is highlighted in blue (line 188-189).

Comment 32: Lines 168 and 169:

Rewrite the sentence for:

“SST positive neurons are present in large concentrations in the intestinal ENS, …” to “SST-positive neurons are present in large concentrations in the intestinal ENS, ...”

Response 32: The sentence has been rewritten as suggested and is highlighted in blue (line 189).

Comment 33: Lines 173 and 174:

Rewrite the sentence for:

“These studies showcase the important part of SST in regulating GI motility.” to “These studies highlight the important role of SST in regulating GI motility.”

Response 33: The sentence has been rewritten as suggested and is highlighted in blue (line 194-195).

Comment 34: Lines 192 to 195:

Rewrite the sentence for:

“HCL presence allows the activation of enzymes involved in digestion, such as pepsin, and promotes the absorption of vitamins and essential nutrients in the digestive tract, while it also has a protective effect against bacteria and other microorganisms [48,49].” to “HCl presence allows the activation of enzymes involved in digestion, such as pepsin, promotes the absorption of vitamins and essential nutrients in the digestive tract, and has a protective effect against bacteria and other microorganisms [48,49].” In this way, the text is presented in a more fluid way.

Response 34: The sentence has been rewritten as suggested and is highlighted in blue (line 213-216).

Comment 35: Line 198:

Rewrite the sentence for:

“Regulation of HCL” to “The regulation of HCl”

Response 35: The sentence has been rewritten as suggested and is highlighted in blue (line 219).

Comment 36: Line 201:

Rewrite the sentence for:

“… HCL ...” to “…. HCl ...”

Response 36: The sentence has been rewritten as suggested and is highlighted in blue (line 222).

Comment 37: Line 202:

Rewrite the sentence for:

“Irritation ...” to “ Activation ...”

Response 37: The sentence has been rewritten as suggested and is highlighted in blue (line 223).

Comment 38: Line 205:

Rewrite the sentence for:

“… gastric acid from parietal cells and histamine from ECL cells.” to “ … gastric acid by parietal cells and histamine by ECL cells.”

Response 38: The sentence has been rewritten as suggested and is highlighted in blue (line 226).

Comment 39: Line 214:

Rewrite the sentence for:

“… the Helicobacter Pylori (HP) (Figure 2.” to “… the Helicobacter Pylori (Figure 2).”

Additionally, HP is not a valid taxonomic abbreviation, so I suggest replacing HP (throughout the section) with H. pylori.

Response 39: The sentence has been rewritten as suggested and is highlighted in yellow (line 236). H.pylori has been used throughout the text instead of HP (lines 254, 270 in yellow).

Comment 40: Lines 225 and 226:

Rewrite the sentence for:

“Loss of these effects results in atrophic gastritis, which is a predisposing factor for gastric malignancies [65].” to “The loss of these effects results in atrophic gastritis, a predisposing factor for gastric malignancies [65].” In this way, the text is presented in a more fluid way.

Response 40: The sentence has been rewritten as suggested and is highlighted in blue (line 247-248).

Comment 41: Line 251:

Rewrite the sentence for:

“…influence …” to “… modulate …”

Response 41: The sentence has been rewritten as suggested and is highlighted in blue (line 273).

Comment 42: Lines 253 to 255:

Rewrite the sentence for:

“CCK secretion is further promoted by the presence of a CCK releasing peptide (CCK-RP) from intestinal cells [68].” to “CCK secretion is further stimulated by CCK-releasing peptide (CCK-RP), which is produced by intestinal cells [68].” In this way, the text is presented in a more fluid way.

Response 42: The sentence has been rewritten as suggested and is highlighted in blue (line 275-277).

Comment 43: Lines 255 and 257:

Rewrite the sentence for:

“By interacting with its receptors (CCKRs), CCK increases gallbladder contraction, stimulates the secretion of pancreatic enzymes and delays gastric emptying [69]. These events are very important for lipid and protein digestion.” to “By binding to its receptors (CCKRs), CCK enhances gallbladder contraction, stimulates pancreatic enzyme secretion, and delays gastric emptying [69]. These processes are essential for lipid and protein digestion.” In this way, the text is presented in a more fluid way.

Response 43: The sentence has been rewritten as suggested and is highlighted in blue (lines 277-279).

Comment 44: Line 258:

Rewrite the sentence for:

“...interacting with CCKRA and causing SST release from D-cells” to “...activating CCKRA, which induces SST release from D-cells”.

Response 44: The sentence has been rewritten as suggested and is highlighted in blue (line 280).

Comment 45: Lines 259 to 262:

Rewrite the sentence for:

“... many studies show it also negatively affects CCK action [71]. Herzig et al. observed decreased secretion of pancreatic enzymes due to inhibition of CCK-RP by SST [72], while Miyasaka et al. found that CCK and CCK-RP secretion was decreased after administering the SSTA octreotide to rats [73].” to “... several studies indicate that it also inhibits CCK activity [71]. Herzig et al. observed reduced pancreatic enzyme secretion due to SST-mediated inhibition of CCK-RP [72]. Similarly, Miyasaka et al. found that administration of the SSTA octreotide to rats decreased both CCK and CCK-RP secretion [73].” In this way, the text is presented in a more fluid way.

Response 45: The sentence has been rewritten as suggested and is highlighted in blue (line 281-284).

Comment 46: Lines 275 to 278:

Rewrite the sentence for:

“SST regulates ghrelin metabolism through different mechanisms. In the CNS, SST interacts with SSTR2 to cause an increase of ghrelin concentration after stress-related suppression of food intake and gastric emptying.” to “SST regulates ghrelin metabolism through multiple mechanisms. In the CNS, SST interacts with SSTR2, leading to increased ghrelin levels following stress-related suppression of food intake and gastric emptying.” In this way, the text is presented in a more fluid way.

Response 46: The sentence has been rewritten as suggested and is highlighted in blue (lines 297-300).

Comment 47: Lines 275 to 278:

Rewrite the sentence for:

“Orgaard et al. observed that the inhibitory effect of GLP-1 on glycagon secretion was diminished after blockage of SSTRs in rats [82].” to “Orgaard et al. observed that GLP1’s inhibitory effect on glucagon secretion was reduced following SSTR blockade in rats [82].” In this way, the text is presented in a more fluid way.

Response 47: The sentence has been rewritten as suggested and is highlighted in blue (line 307-309).

Comment 48: Lines 295, 353, and 499:

Rewrite the sentence for:

“… in vitro ...” to “… in vitro ...”

Response 48: The sentence has been rewritten as suggested and is highlighted in blue (line 317).

Comment 49: Lines 296 to 298:

Rewrite the sentence for:

“SST-SSTRs interaction suppresses adenylate cyclase and alters the permeability of electrolyte channels, such as calcium and potassium channels, on the cytoplasmic membrane [86].” to “SST-SSTR interaction suppresses adenylate cyclase activity and alters the permeability of electrolyte channels, including calcium and potassium channels, in the plasma membrane [86].” In this way, the text is presented in a more fluid way.

Response 49: The sentence has been rewritten as suggested and is highlighted in blue (lines 318-320).

Comment 50: Line 298:

Rewrite the sentence for:

“Expression of Na+/H+ (NHE)…” to “Expression of Na+/H+ (NHE) …”

Response 50: The sentence has been rewritten as suggested and is highlighted in blue (line 320).

Comment 51: Lines 323 to 325:

Rewrite the sentence for:

“The intestinal barrier is a structure mainly comprised of gut microbiota, mucus, epithelial cells and cells of the immune system and their products.” to “The intestinal barrier is primarily composed of gut microbiota, mucus, epithelial cells, immune cells, and their products.” In this way, the text is presented in a more fluid way.

Response 51: The sentence has been rewritten as suggested and is highlighted in blue (lines 345-346).

Comment 52: Lines 328 and 329:

Rewrite the sentence for:

“Some of them, including SST, act on the components of the intestinal layer and play a pivotal role in the preservation of it normal structure and function [83]” to “Some of these peptides, including SST, act on the intestinal barrier components and play a pivotal role in preserving its normal structure and function [83]” In this way, the text is presented in a more fluid way.

Response 52: The sentence has been rewritten as suggested and is highlighted in blue (lines 349-351).

Comment 53: Lines 333 to 335:

Rewrite the sentence for:

“SST exposure and interaction with SSTR5 resulted in increased MUC2 production due to suppression of the Notch-Hes1 pathway [90].” to “SST exposure and interaction with SSTR5 led to increased MUC2 production through the suppression of the Notch-Hes1 pathway [90].” In this way, the text is presented in a more fluid way.

Response 53: The sentence has been rewritten as suggested and is highlighted in blue (lines 355-357).

Comment 54: Lines 338 to 340:

Rewrite the sentence for:

“SST also has a protective effect on the epithelial component of the intestinal barrier. Epithelial cells are connected and their structure is stabilized through a variety of cytoplasmic and transmembrane proteins which form tight junctions (TJs).” to “SST also exerts a protective effect on the epithelial component of the intestinal barrier. Epithelial cells are interconnected, and their structure is stabilized by various cytoplasmic and transmembrane proteins that form tight junctions (TJs).” In this way, the text is presented in a more fluid way.

Response 54: The sentence has been rewritten as suggested and is highlighted in blue (lines 360-362).

Comment 55: Lines 343 to 345:

Rewrite the sentence for:

“In the GI tract, several studies have examined the effect of SST on epithelial cells, and pathways that are involved in preserving barrier integrity and function.” to “Several studies have examined the effects of SST on epithelial cells and its role in preserving barrier integrity and function in the GI tract. In this way, the text is presented in a more fluid way.

Response 55: The sentence has been rewritten as suggested and is highlighted in blue (lines 365-366).

Comment 56: Line 345:

Rewrite the sentence for:

“... ZO-1, ...” to “… Zonula Occludens-1 (ZO-1), ...”

Response 56: The sentence has been rewritten as suggested and is highlighted in blue (line 367).

Comment 57: Lines 349 and 350:

Rewrite the sentence for:

“Signaling via the ERK1/2-MAPK pathway was downregulated by SST in these models [96,97].” to “SST downregulated signaling through the ERK1/2-MAPK pathway in these models [96,97].”

Response 57: The sentence has been rewritten as suggested and is highlighted in blue (lines 371-372).

Comment 58: Line 393:

Rewrite the sentence for:

 “… various medical conditions.” to “… various clinical conditions.”

Response 58: The sentence has been rewritten as suggested and is highlighted in blue (line 415).

Comment 59: Line 396:

Rewrite the sentence for:

“... in the past decades, with few of them, ...” to “... in recent decades, with a few of them, ...”

Response 59: The sentence has been rewritten as suggested and is highlighted in blue (line 418).

Comment 60: Line 398:

Rewrite the sentence for:

“... from ...” to “... by ...”

Response 60: The sentence has been rewritten as suggested and is highlighted in blue (line 420).

Comment 61: Line 411:

Rewrite the sentence for

“…is…” to “…remains…”

Response 61: The sentence has been rewritten as suggested and is highlighted in blue (line 433).

Comment 62: Line 412:

Rewrite the sentence for

“…human…” to “…humans…”

Response 62: The sentence has been rewritten as suggested and is highlighted in blue (line 434).

Comment 63: Lines 414 and 415:

Rewrite the sentence for

“Interaction of octreotide with SSTR2 directly promotes vasoconstriction, while it also alters the effect of other vasoactive peptides.” to “The interaction of octreotide with SSTR2 directly promotes vasoconstriction, while also altering the effect of other vasoactive peptides.” In this way, the text is presented in a more fluid way.

Response 63: The sentence has been rewritten as suggested and is highlighted in blue (lines 436-437).

Comment 64: Line 416:

Rewrite the sentence for:

“… 21 RCTs showed” to “… 21 Randomized Controlled Trials (RCTs).”

Response 64: The sentence has been rewritten as suggested and is highlighted in blue (line 439).

Comment 65: Line 419:

Rewrite the sentence for:

“ESGE guidelines …” to “European Society of Gastrointestinal Endoscopy (ESGE) guidelines ...”

Response 65: The sentence has been rewritten as suggested and is highlighted in blue (lines 441-442).

Comment 66: Line 420:

Rewrite the sentence for:

“...favourable...” to “...favorable...”

Response 66: The sentence has been rewritten as suggested and is highlighted in blue (line 444).

Comment 67: Lines 421 and 422:

Rewrite the sentence for:

“...24 hour to a 72 hour…” to “…24-hour to a 72-hour…”

Response 67: The sentence has been rewritten as suggested and is highlighted in blue (line 445).

Comment 68: Lines 428 and 431:

Rewrite the sentence for:

“... difficult, due to the presence of comorbidities and common recurrence of bleeding in many cases [141]. This leads to increasing requirements for red blood cell and iron transfusion and more frequent hospitalization for these patients.” to “... difficult due to the presence of comorbidities and the common recurrence of bleeding in many cases [141]. This leads to an increasing requirement for red blood cell and iron transfusions, as well as more frequent hospitalizations for these patients.” In this way, the text is presented in a more fluid way.

Response 68: The sentence has been rewritten as suggested and is highlighted in blue (lines 452-455).

Comment 69: Line 432:

Rewrite the sentence for:

“…first line…” to “…the first-line…”

Response 69: The sentence has been rewritten as suggested and is highlighted in blue (line 456).

Comment 70: Lines 434 and 435:

Rewrite the sentence for:

“... making the possibility of untreated lesions higher [143].” to “... increasing the likelihood of untreated lesions [143].”

Response 70: The sentence has been rewritten as suggested and is highlighted in blue (line 459).

Comment 71: Lines 435 and 436:

Rewrite the sentence for:

“... frequent endoscopy procedures is not ideal for both patients and doctors, while health care costs ...” to “... frequent endoscopic procedures is not ideal for either patients or doctors, and healthcare costs...”

Response 71: The sentence has been rewritten as suggested and is highlighted in blue (line 460).

Comment 72: Line 438:

Rewrite the sentence for:

“...aggregation, reduction of intestinal ...” to “... aggregation, reduce intestinal ...”

Response 72: The sentence has been rewritten as suggested and is highlighted in blue (line 462).

Comment 73: Lines 451 and 452:

Rewrite the sentence for:

“NETs are an uncommon form of neoplasms which are typically comprised of neuroendocrine cells.” to “NETs are an uncommon form of neoplasms that are typically composed of neuroendocrine cells.” In this way, the text is presented in a more fluid way.

Response 73: The sentence has been rewritten as suggested and is highlighted in blue (lines 475-476).

Comment 74: Line 453:

Rewrite the sentence for:

“Prognosis of these ...” to “The prognosis for these...”

Response 74: The sentence has been rewritten as suggested and is highlighted in blue (line 477).

Comment 75: Line 454:

Rewrite the sentence for:

“The GI tract is frequent ...” to "The GI tract is a frequent…”

Response 75: The sentence has been rewritten as suggested and is highlighted in blue (line 478).

Comment 76: Line 455:

Rewrite the sentence for:

“ …expressed on… “ to “…expressed in…”

Response 76: The sentence has been rewritten as suggested and is highlighted in blue (line 479).

Comment 77: Lines 461 and 462:

Rewrite the sentence for:

“... symptoms that take place after the direct release of bioactive molecules in the systemic circulation ...” to “... symptoms that occur after the direct release of bioactive molecules into the systemic circulation ...” In this way, the text is presented in a more fluid way.

Response 77: The sentence has been rewritten as suggested and is highlighted in blue (line 486).

Comment 78: Line 464:

Rewrite the sentence for:

 “... syndrome and thus improve quality ...” to “... syndrome, thus improving the quality ...”

Response 78: The sentence has been rewritten as suggested and is highlighted in blue (line 487).

Comment 79: Line 465:

Rewrite the sentence for:

“... first line approach...” to “... first-line approach”

Response 79: The sentence has been rewritten as suggested and is highlighted in blue (line 489).

Comment 80: Line 466:

Rewrite the sentence for:

  “…slowly-growing GI” to “... slowly growing GI”

Response 80: The sentence has been rewritten as suggested and is highlighted in blue (line 491).

Comment 81: Lines 470 and 471:

Rewrite the sentence for:

“... ligands is very useful in the diagnosis and therapy of these patients.” to “... ligands are very useful in the diagnosis and treatment of these patients.”

Response 81: The sentence has been rewritten as suggested and is highlighted in blue (line 495).

Comment 82: Lines 484 and 485:

Rewrite the sentence for:

“... inhibit cell proliferation and migration, and induce apoptosis [28].” to “... inhibit cell proliferation, migration, and induce apoptosis [28].”

Response 82: The sentence has been rewritten as suggested and is highlighted in blue (lines 509-510).

Comment 83: Line 493:

Rewrite the sentence for:

“... with poor CRC ...” to “... with a poor CRC ...”

Response 83: The sentence has been rewritten as suggested and is highlighted in blue (line 518).

Comment 84: Line 518:

“variatric” should likely be “bariatric” (Check this out).

Response 84: The sentence has been rewritten as suggested and is highlighted in blue (line 543).

Comment 85: Line 520:

Rewrite the sentence for:

“... complain of symptoms ...” to “... experience symptoms ...”

Response 85: The sentence has been rewritten as suggested and is highlighted in blue (line 545).

Comment 86: Line 521:

Rewrite the sentence for:

“... after meal ingestion.” to “... after eating a meal.”

Response 86: The sentence has been rewritten as suggested and is highlighted in blue (line 546).

Comment 87: Lines 529 and 530:

Rewrite the sentence for:

 “... small intestine and affect hormone and electrolyte distribution, they have been tested in the treatment of dumping syndrome [125].” to “... small intestine, and affect hormone and electrolyte distribution. They have been tested in the treatment of dumping syndrome [125].” In this way, the text is presented in a more fluid way.

Response 87: The sentence has been rewritten as suggested and is highlighted in blue (lines 553-555).

Comment 88: Lines 549 and 551:

Rewrite the sentence for:

“Suppression of gastrointestinal motility and secretion of pancreatic enzymes by SST and SSTAs make them a possible treatment option for these patients [164].” to “Suppression of gastrointestinal motility and pancreatic enzyme secretion by SST and SSTAs makes them a possible treatment option for these patients [164].” In this way, the text is presented in a more fluid way.

Response 88: The sentence has been rewritten as suggested and is highlighted in blue (lines 574-576).

Comment 89: Lines 554 and 555:

Rewrite the sentence for:

“...  evidence for general SST analogue application as anti-diarrheal agents in clinical practice [86,164].” to “… evidence for the general use of SST analogues as anti-diarrheal agents in clinical practice [86,164].”

Response 89: The sentence has been rewritten as suggested and is highlighted in blue (lines 579-580).

Comment 90: Lines 571 and 572:

Rewrite the sentence for:

“...  many different tissues and cause a variety of effects.” to “... various tissues and produce a range of effects.”

Response 90: The sentence has been rewritten as suggested and is highlighted in blue (lines 616-617).

Comment 91: Lines 579 and 582:

Rewrite the sentence for:

“The more frequent use of exciting new therapeutic techniques, including PPRT, and the positive effect of SSTAs in different conditions are well documented, however future studies are needed to determine which patients will mostly benefit from their off-label use, and thus improve patient care.” to “The increasing use of promising new therapeutic techniques, including PPRT, and the positive effects of SSTAs in various conditions are well-documented. However, future studies are needed to determine which patients will most benefit from their off-label use, thus improving patient care.”

Response 91: The sentence has been rewritten as suggested and is highlighted in blue (lines 624-727).

Reviewer 2 Report

Comments and Suggestions for Authors

The authors in this review manuscript titled ,, The Role of Somatostatin in the Gastrointestinal Tract” conducted an analysis based on the bibliographic study regarding the importance of somatostatin and its role in relation to the gastrointestinal tract. The authors describe in detail somatostatin production and secretion, gut receptors for somatostatin, effects of somatostatin on different gastrointestinal functions, the use of somatostatin analogues as treatment in different gastrointestinal diseases. The references are numerous and relevant to the topic. The figures and table complete and highlight the explanations better. The conclusions support the discussions.

The manuscript has some issues that I invite the authors to answer:

Because the analysis is based on studying references, it is necessary to add how you selected the articles analyzed: what was the databases in which the articles were searched? in what language were the articles searched? what was the inclusion and exclusion list?  how many articles were evaluated and how many were rejected from this analysis?

A few paragraphs should be written about the limitations of this review and possible new research or analysis.

Bacterial names should be written in italics (ex line 214).

Author Response

We would like to thank you for helping us improve our manuscript. Here are our responses to your comments.

Comment 1: Because the analysis is based on studying references, it is necessary to add how you selected the articles analyzed: what was the databases in which the articles were searched? in what language were the articles searched? what was the inclusion and exclusion list?  how many articles were evaluated and how many were rejected from this analysis?

Response 1: A section titled "2. Methodology" has been added to the manuscript. As this review was not systematic, we have not kept a record of how many articles were evaluated in total and how many were eventually rejected. We have added this as a limitation of our review in the section titled "6. Study Limitations", which has also been added to the manuscript and is highlighted in yellow. 

Comment 2: A few paragraphs should be written about the limitations of this review and possible new research or analysis. 

Response 2: A new section titled "6. Study Limitations "and a paragraph titled5.7. Limitations of somatostatin analogues in clinical practicehave been added to the manuscript and are highlighted in yellow.

Comment 3: Bacterial names should be written in italics (ex line 214).

Response 3: All bacterial names have been rewritten in italics and are highlighted in yellow.

Reviewer 3 Report

Comments and Suggestions for Authors

The manuscript offers a comprehensive and well-organized overview of the roles of gastrointestinal peptides in regulating glucose homeostasis and energy balance, with a particular emphasis on their physiological functions and therapeutic potential. The authors have successfully synthesized a substantial body of literature, presenting a coherent narrative that connects mechanistic insights with translational applications in metabolic disorders, especially type 2 diabetes and obesity. The discussion of recent therapeutic developments—particularly GLP-1 receptor agonists and dual/triple agonists—is timely and highly relevant given the rapid advancements in the field of metabolic disease treatment. I recommend this review for publication after addressing the following minor issues:

  1. While the review is informative, it would benefit from more critical analysis or discussion of conflicting findings, especially in areas where the evidence is inconclusive or controversial (e.g., the role of GIP in obesity).
  2. The authors may consider briefly introducing emerging or less-studied peptides that could represent promising directions for future research in glucose metabolism and appetite regulation.
  3. Consistent use of terminology and abbreviations is encouraged throughout the manuscript (e.g., uniformly using either “GLP-1 receptor agonists” or “GLP-1RAs”).
  4. Minor typographical and grammatical corrections are recommended to improve clarity and readability in certain sections.

Author Response

We would like to thank you for helping us improve our manuscript. Here are our responses to your comments.

Comment 1: While the review is informative, it would benefit from more critical analysis or discussion of conflicting findings, especially in areas where the evidence is inconclusive or controversial (e.g., the role of GIP in obesity).

Response 1: We have discussed conflicting findings and inconclusive evidence regarding the role of somatostatin and its analogues in the gastrointestinal tract (e.g. off label use of somatostatin analogues for dumping syndrome due to limited effect on the delayed phase and many side effects, the effect of somatostatin on the gut immune system). Although the study of different peptides secreted in the gut and their part in different diseases has provided interesting but at many cases conflicting evidence, we feel that expanding on that evidence does not align with the purpose of our analysis. 

Comment 2: The authors may consider briefly introducing emerging or less-studied peptides that could represent promising directions for future research in glucose metabolism and appetite regulation.

Response 2: Peptides and receptors that show promise in the regulation of glucose metabolism and appetite regulation are a very interesting topic of ongoing investigation. However, as we have decided to focus our article on the functions of somatostatin and its analogues in the gastrointestinal tract, we feel that expanding on that topic does not align with the purpose of our analysis.

Comment 3: Consistent use of terminology and abbreviations is encouraged throughout the manuscript (e.g., uniformly using either “GLP-1 receptor agonists” or “GLP-1RAs”).

Response 3: Consistent use of terminology and abbreviations has been used throughout the text (e.g. somatostatin- SST, SSTR1-STTR5, H. pylori).

Comment 4: Minor typographical and grammatical corrections are recommended to improve clarity and readability in certain sections.

Response 4: A thorough re-reading of the text was done to further assess its readability. We made typographical and grammatical corrections throughout our manuscript to improve its clarity.

Reviewer 4 Report

Comments and Suggestions for Authors

This is a review that describes the role of somatostatin in the digestive tract. Somatostatin is a hormone secreted by the pancreas that inhibits the secretion of insulin and glucagon, as well as gastrin, cholecystokinin, and secretin. The relationship between somatostatin and the pancreas should be explained in more detail.

Author Response

Thank you for taking the time to help us improve our manuscript!

Comment 1: This is a review that describes the role of somatostatin in the digestive tract. Somatostatin is a hormone secreted by the pancreas that inhibits the secretion of insulin and glucagon, as well as gastrin, cholecystokinin, and secretin. The relationship between somatostatin and the pancreas should be explained in more detail.

Response 1: The effect of somatostatin on pancreatic cells and its role on glucose metabolism is a very important topic of ongoing investigation. However, these effects have been recently analyzed in the literature. Moreover, we wanted to focus our analysis on the effects of somatostatin on the gastrointestinal tract and not its accessory organs and we felt that expanding on the relationship between somatostatin and the pancreas did not align with our aim. We have added comments in the first paragraph of section 4.1 (lines 163-168 highlighted in yellow) and written "The important role of SST in the regulation of insulin and glucagon secretion has also been demonstrated in both human and animal models [37-39]. As the interaction between different pancreatic cells and the effects of SST in glucose metabolism have been recently analyzed in the literature, viewers are encouraged to extend their reading on other recent articles regarding this topic [40,41]." We have also added some references on the topic and encouraged viewers to further read on the subject.